# Curly Flow Matching for Learning Non-gradient Field Dynamics

**Katarina Petrović**[1],[*] **Lazar Atanackovic**[2],[3],[4], **Viggo Moro**[1], **Kacper Kapuśniak**[1],
**İsmail İlkan Ceylan**[5],[6],[1], **Michael Bronstein**[1],[6], **Avishek Joey Bose**[1],[7],[†] **Alexander Tong**[6],[7],[8],[†]

[1]University of Oxford, [2]Broad Institute of MIT and Harvard, [3]University of Toronto,
[4]Vector Institute, [5]TU Wien, [6]AITHYRA, [7]Mila – Quebec AI Institute, [8]Université de Montréal

## Abstract

Modeling the transport dynamics of natural processes from population-level observations is a ubiquitous problem in the natural sciences. Such models rely on key assumptions about the underlying process in order to enable faithful learning of governing dynamics that mimic the actual system behavior. The de facto assumption in current approaches relies on the principle of least action that results in gradient field dynamics and leads to trajectories minimizing an energy functional between two probability measures. However, many real-world systems, such as cell cycles in single-cell RNA, are known to exhibit non-gradient, periodic behavior, which *fundamentally* cannot be captured by current state-of-the-art methods such as flow and bridge matching. In this paper, we introduce CURLY FLOW MATCHING (CURLY-FM), a novel approach that is capable of learning non-gradient field dynamics by designing and solving a Schrödinger bridge problem with a non-zero drift reference process—in stark contrast to typical zero-drift reference processes—which is constructed using *inferred velocities* in addition to population snapshot data. We showcase CURLY-FM by solving the trajectory inference problems for single cells, computational fluid dynamics, and ocean currents with approximate velocities. We demonstrate that CURLY-FM can learn trajectories that better match both the reference process and population marginals. CURLY-FM expands flow matching models beyond the modeling of populations and towards the modeling of known periodic behavior in physical systems. Our code repository is accessible at: https://github.com/kpetrovicc/curly-flow-matching.git.

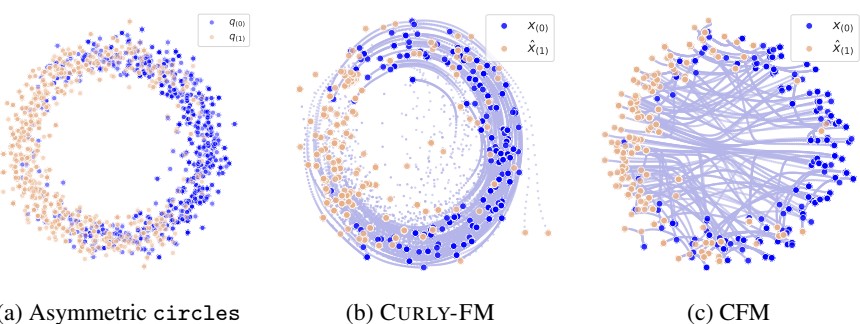

| (a) Asymmetric `circles` | (b) CURLY-FM | (c) CFM |

Figure 1: Particle trajectories generated between samples drawn from asymmetric `circles` distribution at $t = 0$ and $t = 1$ and respective to underlying reference velocity field $f_t(x_t)$. Traditional flow-based models such as OT-CFM and CFM *cannot* capture cyclical patterns in physical systems. CURLY-FM is capable of *learning non-gradient field dynamics* behavior in the underlying data.

---

[*]Correspondence to: {katarina.petrovic}@cs.ox.ac.uk

[†]Equal advising

39th Conference on Neural Information Processing Systems (NeurIPS 2025).

# 1 Introduction

Understanding the temporal evolution of multi-body systems remains a central challenge across many applications in life sciences [Lange et al., 2024, Pan and Zhang, 2024], as such systems are often characterized by complex dynamics governing their evolutionary behavior. For example, biological tissues are complex systems that evolve through tissue differentiation characterized by cell divisions and deaths as well as morphological changes. Learning evolutionary behavior in such systems can be formulated as a *trajectory inference* problem [Hashimoto et al., 2016, Lavenant et al., 2024], where the goal is to recover full trajectories of particles given partial and noisy population-level snapshots.

The dominant paradigm in solving the trajectory inference problems in scientific applications involves leveraging tools from computational optimal transport (OT) [Peyré and Cuturi, 2019] to learn neural dynamical systems, e.g. NeuralODE [Chen et al., 2018], such that sampled trajectories under the model optimize a notion of likeliness of being observed [Bunne et al., 2024b]. For instance, in single-cell trajectory inference, such methods broadly follow a pipeline that first infers "optimal" cell trajectories that follow the gradient of some potential function (often termed a Waddington Landscape [Waddington, 1942]), before searching for important regulators of a biological process in development [Schiebinger et al., 2019, Shahan et al., 2022] or disease [Tong et al., 2023, Klein et al., 2025]. Despite the ability to produce approximately optimal trajectories w.r.t. the energy landscape, current methods are limited in their ability to model only gradient-field dynamics. Consequently, trajectories inferred under the model are not realistic and fail to model crucial system dynamics such as periodic behavior that arises in natural systems, e.g., cell cycling [Riba et al., 2022] where periodic behavior is observed at the scale of months, days, hours, and minutes. These behaviors cannot be addressed by current OT-based methods, as gradient field dynamics cannot capture periodic behavior.

**Present work**. In this paper, we tackle the modeling of systems governed by non-gradient field dynamics. We introduce CURLY FLOW MATCHING (CURLY-FM), a novel approach for learning non-gradient field dynamics by solving a Schrödinger bridge problem with a designed reference process that induces the learning of periodic behavior. Specifically, we consider reference processes with non-zero drift—in stark contrast to zero drift processes in approaches such as Diffusion Schrödinger Bridges [De Bortoli et al., 2021b, Shi et al., 2024]. Such a modification elevates the established (entropic optimal) mass transport problem to a new class of problems that require matching the reference drift while also transporting mass between time marginals associated with observations. As a result, solutions to this Schrödinger bridge problem are capable of learning non-gradient field dynamics and exhibit behaviors such as periodicity as found in cell cycling.

In addition to conventional population snapshot data, we design CURLY-FM by leveraging approximate velocity information that is used to construct the drift of a reference process. Consequently, to model periodic dynamics CURLY-FM solves the Schrödinger bridge problem by decomposing it into a two-stage algorithm. The first stage learns a neural path interpolant by regressing against the drift of our constructed reference process. Unlike straight paths in optimal transform conditional flow matching trajectories, the neural path interpolant exhibits cyclic behavior due to the optimization objective of matching a constructed reference drift. In stage two, we learn, in a simulation-free manner, to construct the generative process that solves the mass transport problem as a mixture of conditional bridges built using optimal transport-based couplings that minimize the length of the velocity field of the neural path interpolant. The combination of our two-stage approach enables CURLY-FM to learn dynamics that *do not* affect the population data, but do affect individual particles, which include observed periodic non-gradient field dynamics, unlike other methods (see table 1).

We instantiate CURLY-FM on modeling a suite of problems in natural sciences that exhibit known non-gradient field dynamics, such as cell cycle systems found in scRNA-seq data with RNA-velocity [La Manno et al., 2018], ocean currents, and computational fluid dynamics for PDEs simulated using the Lagrangian particle discretizations [Toshev et al., 2023]. In each case, the system under consideration is under the influence of a *known* reference process, e.g., RNA-velocity gives an approximation of the instantaneous velocity of a cell, which must be adhered to when solving the trajectory inference problem. More precisely, using the reference process drift and population snapshots CURLY-FM solves the Schrödinger bridge problem by searching for a bridge that matches the reference process as closely as possible, but also matches the marginal distributions at both timepoints of the system dynamics. Consequently, we show that previous flow matching approaches fail to model this cyclic behavior as they are not able to take advantage of the additional reference process information. Furthermore, while there exist simulation-based methods that are in principle able to learn the correct dynamics [Tong et al., 2020], we show that in practice CURLY-FM performs

significantly better both in terms of accuracy to match the reference drift—enabling modeling of cyclic behavior (c.f. fig. 1)—while being computationally cheaper due to its simulation-free training nature.

## 2    Background and preliminaries

Given two distributions $\rho_0$ and $\rho_1$, the *distributional matching problem* seeks to find a push-forward map $\psi : \mathbb{R}^d \to \mathbb{R}^d$ that transports the initial distribution to the desired endpoint, $\rho_1 = [\psi]_\#(\rho_0)$. Such a problem setup is pervasive in many areas of machine learning and notably encompasses the standard generative modeling and optimal transport settings [De Bortoli et al., 2021a, Peyré et al., 2019]. In this paper, we consider the setting where each distribution $\rho_0$ and $\rho_1$ is an empirical distribution that is accessible through a dataset of observations $\{x_0^i\}_{i=1}^N \sim \rho_0(x_0)$ and $\{x_1^j\}_{j=1}^N \sim \rho_1(x_1)$. Thus the modeling task is to learn the (approximate) transport map $\psi$.

### 2.1    Continuous normalizing flows and flow matching

One common choice for modeling $\psi$ is as a deterministic dynamic system with a time-dependent generator $\psi_t : [0,1] \times \mathbb{R}^d \to \mathbb{R}^d$. The solution to this dynamical system is an ordinary differential equation (ODE) and the learned transport map is known as a *continuous normalizing flow* (CNF). A CNF is a time-indexed neural transport map $\psi_t$, for all time $t \in [0,1]$, that is trained to push forward samples from prior $\mu_0$ to a desired target $\mu_1$. Specifically, a CNF models the ODE $\frac{d}{dt}\psi_t(x) = f_t(\psi_t(x))$ with initial conditions $\psi_0(x_0) = x_0$ and $f_t : [0,1] \times \mathbb{R}^d \to \mathbb{R}^d$ being the time-dependent vector field associated with the ODE and transports samples from $\mu_0 \to \mu_1$.

The most scalable way to train CNFs is to utilize a *simulation-free* training objective which regresses a learned neural vector field $v_{t,\theta}(x_t) : [0,1] \times \mathbb{R}^d \to \mathbb{R}^d$ to the desired target vector field $f_t(x_t)$ for all time. This technique is commonly known as flow-matching [Liu, 2022, Albergo and Vanden-Eijnden, 2023, Lipman et al., 2023, Tong et al., 2024a] and has the neural transport map $\psi_{t,\theta}$ which is obtained through a neural differential equation [Chen et al., 2018] $\frac{d}{dt}\psi_{t,\theta}(x) = v_{t,\theta}(\psi_{t,\theta}(x))$. Specifically, flow-matching regresses $v_{t,\theta}(x_t)$ to the target *conditional* vector field $f_t(x_t|z)$ associated to the target flow $\psi_t(x_t|z)$. We say that this conditional vector field $f_t(x_t|z)$, *generates* the target density $\mu_1(x_1)$ by interpolating along the probability path $\mu_t(x_t|z)$ in time. We often do not have closed-form access to the generating marginal vector field $f_t(x_t)$. Still, with conditioning, e.g., $z = (x_0, x_1)$, we can obtain a simple analytic expression of a conditional vector field that achieves the same goals. The conditional flow-matching (CFM) objective can then be stated as a simple simulation-free regression,

$$\mathcal{L}_{\text{CFM}}(\theta) = \mathbb{E}_{t,q(z),\mu_t(x_t|z)}\|v_{t,\theta}(t,x_t) - f_t(x_t|z)\|_2^2. \tag{1}$$

The conditioning distribution $q(z)$ can be chosen from any valid coupling, for instance, the independent coupling $q(z) = \mu_0(x_0)\mu_1(x_1)$. To generate samples and their corresponding log density according to the CNF, we may solve the following flow ODE numerically with initial conditions $x_0 = \psi_0(x_0)$ and $c = \log\mu_0(x_0)$, which is the log density under the prior:

$$\frac{d}{dt}\begin{bmatrix}\psi_{t,\theta}(x_t)\\\log\mu_t(x_t)\end{bmatrix} = \begin{bmatrix}v_{t,\theta}(t,x_t)\\-\nabla\cdot v_{t,\theta}(t,x_t)\end{bmatrix}. \tag{2}$$

In the next section, we outline a different methodology to build a transport map leveraging *stochastic* dynamics. This allows us to frame the mass transport problem as a Schrödinger bridge, which is well suited to modeling noisy measurements in applications such as single-cell evolution.

## 3    Schrödinger Bridge with non-zero reference field

The complex nature of particle dynamics can be captured as a mass transport problem under a prescribed reference process. Specifically, we model particle evolution using a parametrized stochastic differential equation (SDE), with drift $v_{t,\theta} : [0,1] \times \mathbb{R}^d \to \mathbb{R}^d$, diffusion coefficient $g_t > 0$:

$$d\mathbf{X}_t = v_{t,\theta}(\mathbf{X}_t)\,dt + g_t d\mathbf{B}_t, \quad \mathbf{X}_0 \sim \mu_0, \mathbf{X}_1 \sim \mu_1, \tag{3}$$

where $\mathbf{B}_t$ is a standard Brownian motion and by convention time $t \in [0,1]$ flows from $t = 0$ to $t = 1$ such that marginal distribution at the endpoints are $\rho_0$ and $\rho_1$. These endpoints are provided as empirical distributions and represent endpoint observations along a transport trajectory. The SDE in eq. (3) induces a path measure in the space of Markov path measures $(\mathbb{P}_{t,\theta})_{t\in[0,1]} \in \mathcal{P}(C[0,1], \mathbb{R}^d)$

such that the marginal density $p_t$ evolves according to the following Fokker-Plank equation:

$$\frac{\partial p}{\partial t} = -\nabla \cdot (v_{t,\theta}(\mathbf{X}_t), p_t(\mathbf{X}_t)) + \frac{g_t^2}{2} \Delta p_t(\mathbf{X}_t), \quad p_0 = \rho_0, p_1 = \rho_1. \tag{4}$$

In addition, our modeling of particle dynamics is informed by a reference process which is defined by the following SDE with corresponding drift $f_t : \mathbb{R}^d \to \mathbb{R}^d$ and diffusion coefficient $g_t > 0$:

$$d\mathbf{X}_t = f_t(\mathbf{X}_t)\, dt + g_t d\mathbf{B}_t. \tag{5}$$

We denote the induced path measure of eq. (5) as $(\mathbb{Q}_t)_{t \in [0,1]}$.

Note further that we assume the diffusion coefficient, $g_t$, to be the same for both processes $\mathbb{P}_t$ and $\mathbb{Q}_t$ to simplify the setting and facilitate easier exposition of the setting considered in this paper.

**Schrödinger bridge with zero-drift**. We now state the Schrödinger bridge problem, which finds an optimal path measure $\mathbb{P}^*$ that is the solution to the following KL-divergence minimization problem:

$$\mathbb{P}^* = \underset{\theta}{\arg\min}\, [\text{KL}\,(\mathbb{P}_\theta || \mathbb{Q}) : \mathbb{P}_0 = \mu_0, \mathbb{P}_1 = \mu_1] \tag{6}$$

In settings where eq. (5) is zero-drift and with constant diffusion coefficient—i.e. $d\mathbf{X}_t = g_t \mathbf{B}_t$—the Schrödinger bridge problem [Schrödinger, 1932] devolves into the *Diffusion Schrödinger Bridge* problem [De Bortoli et al., 2021a, Bunne et al., 2023]. In this special case, the Schrödinger bridge problem admits a unique solution and is linked to the entropic optimal transport plan through the seminal result of Föllmer [1988]. Specifically, $\mathbb{P}^*$ is a mixture of conditional Brownian bridges $\mathbb{Q}_t(\cdot|x_0, x_1)$ weighted by the entropic OT-plan $\pi^* \in \Pi(\rho_0 \otimes \rho_1)$ which is a valid coupling in the product measure $\rho_0 \otimes \rho_1$, in other words $\int \pi(x_0, \cdot) = \mu_0(x_0), \int \pi(\cdot, x_1) = \mu(x_1)$,

$$\mathbb{P}^* = \int \mathbb{Q}_t(\cdot|x_0, x_1)\, d\pi^*(x_0, x_1) \tag{7}$$

$$\pi^*(\mu_0, \mu_1) = \underset{\pi \in \Pi(\mu_0 \otimes \mu_1)}{\arg\min} \int c(x_0, x_1) d\pi(x_0, x_1) + 2\sigma^2 \text{KL}(\pi || \rho_0 \otimes \rho_1). \tag{8}$$

This holds when $g_t = \sigma$ for some constant $\sigma$ (i.e. $g_t$ is "fixed"), which assume for the remainder of this work. In this case, operationally, the conditional Brownian bridges take the form of a Normal distribution $\mathbb{Q}_t(\cdot|x_0, x_1) = \mathcal{N}(x_t; tx_1 + (1-t)x_0, t(1-t)\sigma^2)$ with the mean given as an interpolation between two endpoints. Furthermore, when $\sigma \to 0$, the entropic OT problem reduces to the regular OT problem. We note that this Schrödinger bridge problem can be reinterpreted as a stochastic optimal control problem where the control cost is the drift $v_{t,\theta}$. That is the stochastic optimal control perspective minimizes *average kinetic energy*[3] of the learned process which leads to the following optimization problem:

$$v_\theta^* = \left\{ \min_\theta \int \mathbb{E}_{\mathbb{P}_t} \left[ \frac{1}{2} \|v_{t,\theta}(\mathbf{X}_t)\|_2^2 \right] dt : d\mathbf{X}_t = v_{t,\theta}(\mathbf{X}_t)\, dt + g_t d\mathbf{B}_t, \mathbb{P}_0 = \rho_0, \mathbb{P}_1 = \rho_1 \right\}. \tag{9}$$

We approximate $\mathbb{P}^*$ using mini-batch OT [Fatras et al., 2020, 2021] and simulation-free matching algorithms [Tong et al., 2024a,b, Pooladian et al., 2023], iterative proportional and Markov fitting [De Bortoli et al., 2021a, Shi et al., 2024], and generalized Schrödinger bridge matching [Liu et al., 2023a].

### 3.1 Schrödinger Bridges with non-zero drift

We now consider the more general case where the drift of the reference process $\mathbb{Q}$ is non-zero. In this case, existing computational approaches, which rely on gradient field dynamics, no longer apply. As detailed in table 1, we consider the case of a constant $g_t > 0$ and tailor our approach to small $g_t$ as previous work has shown this to be preferred empirically [Tong et al.,

Table 1: Overview of the properties of different approaches.

| Method | $\mathbb{P}_\theta$ | $\mathbb{Q}$ | | Models Curl |
|---|---|---|---|---|
| | $v_t$ | $f_t$ | $g_t$ | |
| DSBM | ✓ | ✗ | fixed | ✗ |
| OT-CFM | ✓ | ✗ | $\lim_{g_t \to 0}$ | ✗ |
| GSBM | ✓ | ✓ | learned | ✗ |
| CURLY-FM (ours) | ✓ | ✓ | fixed | ✓ |

2024b]. While it is possible to also learn $g_t$ as in GSBM [Liu et al., 2023a], this is computationally expensive; in this work, we focus on the setting of fixed or small $g_t$ and vectorfields with Curl, as motivated by applications in our experiments section 4. In this case, we can approximate the

---

[3]Schrödinger bridges minimize the relative entropy w.r.t. to $\mathbb{Q}$ and kinetic energy in the deterministic case.

marginal process using a mixture of conditional bridges, albeit not necessarily Brownian. Specifically, we consider the case modeling the marginal process $\mathbb{P}_t = \int \mathbb{Q}_t(\cdot|x_0, x_1)d\pi(x_0, x_1)$, where we use $\mathbb{Q}_t(\cdot|x_0, x_1)$ to denote the stochastic bridge pinned at $x_0, x_1$ at times 0 and 1 respectively and $\pi$ is a valid coupling. With this decomposition, it remains to specify the parameterization of $\mathbb{Q}_t(\cdot|x_0, x_1)$ and $\pi(x_0, x_1)$. As a modeling choice, we parameterize $\mathbb{Q}_t(\cdot|x_0, x_1)$ using a mixture of Brownian bridges with learnable parameters $\eta$:

$$\mathbb{Q}_{\eta,t} = f_{t,\eta}(x_{t,\eta}|x_0, x_1)dt + g_t d\mathbf{B}_t, \tag{10}$$

with the idea that,

$$f_\eta^* = \left\{ \min_\eta \int \mathbb{E}_{\mathbb{P}_t} \left[ \frac{1}{2} \|f_{t,\eta}(x_t) - f_t(x_t)\|_2^2 \right] : d\mathbf{X}_t = f_{t,\eta}(\mathbf{X}_t) dt + g_t d\mathbf{B}_t, \mathbb{Q}_0 = \rho_0, \mathbb{Q}_1 = \rho_1 \right\}.$$

We approximate the mean of $\mathbb{Q}_t$ by designing a neural path interpolant $\varphi_{t,\eta}$ with parameters $\eta$:

$$\mu_{t,\eta} := \int_0^t f_{s,\eta}(\mu_s)ds = tx_1 + (1 - t)x_0 + t(1 - t)\varphi_{t,\eta}(x_0, x_1). \tag{11}$$

We optimize $\varphi_{t,\eta}$ by minimizing the following simulation-free objective of the relative kinetic energy:

$$\mathcal{L}(\eta) = \mathbb{E}_{t\sim\mathcal{U}[0,1],x_0\sim\rho_0,x_1\sim\rho_1} \left[ \left\| \frac{\partial \mu_{t,\eta}}{\partial t} - f_t(\mu_{t,\eta}) \right\|_2^2 \right], \quad f_t(\mu_{t,\eta}) = \kappa(\mu_{t,\eta}, x_0)f_0(x_0).$$

Here $\kappa_t(\mu_{t,\eta}, x_0)$ is any smooth function, e.g. a nearest neighbor based distance kernel $\kappa_t(\mu_{t,\eta}, x_0) = \|\mu_{t,\eta} - x_0^i\|_2 / \sum_i^N \|\mu_{t,\eta} - x_0^i\|_2$. Then we use a kernel definition of $f_t$ based on data as it is uncommon to have access to $f_t$ for the practical applications we consider. We further discuss our assumptions for $\kappa_t$ in the appendix §G.4. We also note $\partial \mu_{t,\eta}/\partial t$ can be computed using automatic differentiation:

$$\frac{\partial \mu_{t,\eta}}{\partial t} = f_{t,\eta}(\mu_t) = x_1 - x_0 + t(1 - t)\frac{\partial \varphi_{t,\eta}}{\partial t}(x_0, x_1) + (1 - 2t)\varphi_{t,\eta}(x_0, x_1). \tag{12}$$

The pseudocode for learning the neural path is presented in algorithm 1. To approximate $\mathbb{P}_t$ we next learn to approximate the optimal mixture of conditional bridges $\mathbb{P}_t^* = \mathbb{E}_{x_0,x_1\sim\pi^*(x_0,x_1)} [x_{t,\eta}]$. However, this necessitates the feasibility of computing the OT-plan, which is defined below:

$$\pi^*(\mu_0, \mu_1) = \operatorname*{argmin}_{\pi\in\Pi(\mu_0\otimes\mu_1)} \int c(x_0, x_1)d\pi(x_0, x_1), \quad c(x_0, x_1) = \int_0^1 \left\| \frac{\partial \mu_{t,\eta}}{\partial t} - f_t(\mu_{t,\eta}) \right\|_2^2 dt$$

$$\text{s.t.} \int \pi(x_0, \cdot) = \rho_0(x_0), \int \pi(\cdot, x_1) = \rho(x_1).$$

Indeed, this optimal transport cost $c(x_0, x_1)$ can be computed through simulating the entire trajectory. However, we opt for using a stochastic estimator of the cost with $K$ samples:

$$c(x_0, x_1) = \mathbb{E}_{t\sim\mathcal{U}[0,1]} \left[ \left\| \frac{\partial x_{t,\eta}}{\partial t} - f_t(x_{t,\eta}) \right\|_2^2 \right] = \frac{1}{K} \sum_i^K \left\| \frac{\partial x_{t,\eta}^i}{\partial t} - f_t(x_{t,\eta}^i) \right\|_2^2. \tag{13}$$

This cost ensures that we choose a coupling which minimizes the total cost in eq. (10). We highlight that the optimal plan $\pi^*$ is intractable and we instead use a biased minibatch approximation of the plan (see §G.3). We use the cost in eq. (13) to estimate a transport plan $\pi(x_0, x_1)$ to construct the approximated mixture of conditional bridges $\mathbb{P}_t$ which is needed to learn the drift $v_{t,\theta}(x_t)$ of eq. (3),

$$\mathcal{L}_{\text{flow}}(\theta) = \mathbb{E}_{t\sim\mathcal{U}[0,1],(x_0,x_1)\sim\pi(x_0,x_1)} \left[ \frac{1}{2} \left\| v_{t,\theta}(x_{t,\eta}) - \left( \frac{\partial \mu_{t,\eta}}{\partial t} \right) .\texttt{detach()} \right\|_2^2 \right].$$

The procedure for this marginal (flow) matching objective is presented in algorithm 2. In the case that $g_t := \sigma > 0$, for a constant $\sigma$, we also need to learn the marginal score $s_{t,\theta} \approx \nabla \log p_t(x_t)$. This can be learned using a conditional score matching objective where $\lambda_t = 2\sqrt{t(1-t)}/\sigma$ and

$x_{t,\eta} = \mu_{t,\eta} + \sigma\epsilon\sqrt{t(1-t)}$ with $\epsilon \sim \mathcal{N}(0,1)$ [Tong et al., 2024b],

$$\mathcal{L}_{\text{score}}(\theta) = \mathbb{E}_{t\sim\mathcal{U}[0,1],(x_0,x_1)\sim\pi(x_0,x_1)}\left[\frac{1}{2}\left\|\lambda_t s_{t,\theta}(x_{t,\eta}) + \epsilon\right\|_2^2\right].$$

In totality, the combined loss is given by thus $\mathcal{L}(\theta) = \mathcal{L}_{\text{flow}}(\theta) + \mathcal{L}_{\text{score}}(\theta)$.

---

**Algorithm 1** Training algorithm for neural path interpolant network

**Require:** Marginals $\rho_0(x_0)$ and $\rho_1(x_1)$, network $\varphi_\eta$, reference drift $f_{t,\eta}$
1: **while** Training **do**
2:     Sample $(x_0, x_1, t) \sim \rho_0(x_0)\rho_1(x_1)\mathcal{U}(0,1)$
3:     $\mu_{t,\eta} = (1-t)x_0 + tx_1 + t(1-t)\varphi_{t,\eta}(x_0,x_1)$
4:     $\frac{\partial\mu_{t,\eta}}{\partial t} = x_1 - x_0 + t(1-t)\frac{\partial\varphi_{t,\eta}(x_0,x_1)}{\partial t} + (1-2t)\varphi_{t,\eta}(x_0,x_1)$
5:     $\mathcal{L}(\eta) = \left\|\frac{\partial\mu_{t,\eta}}{\partial t} - f_{t,\eta}(\mu_{t,\eta})\right\|_2^2$
6:     $\eta \leftarrow \text{Update}(\eta, \nabla_\eta\mathcal{L}(\eta))$
    **return** $\varphi_{t,\eta}$

**Algorithm 2** Marginal Score and Flow Matching

**Require:** Marginals $\rho_0(x_0)$ and $\rho_1(x_1)$, network $\varphi_\eta$, vector field network $v_{t,\theta}$.
1: **while** Training **do**
2:     Sample $(x_0, x_1, t_{ij}) \sim \rho_0(x_0)\rho_1(x_1)\mathcal{U}(0,1)$
3:     $C_\eta^{ij}(x_0^i, x_1^j) = \mathbb{E}_t\left[\left\|\frac{\partial\mu_{t,\eta}}{\partial t} - f_{t,\eta}(\mu_{t,\eta})\right\|_2^2\right]$
4:     $x_0, x_1 \sim \pi(x_0, x_1) \leftarrow \text{OT}(x_0, x_1, C_\eta)$
5:     $t \sim \mathcal{U}(0,1), \epsilon \sim \mathcal{N}(0,1)$
6:     $\mathcal{L}(\theta) = \mathcal{L}_{\text{flow}}(\theta) + \mathcal{L}_{\text{score}}(\theta)$
7:     $\theta \leftarrow \text{Update}(\theta, \nabla_\theta\mathcal{L}(\theta))$
    **return** $v_\theta$

---

**Remark 1.** *We highlight that while $\mathcal{L}(\theta)$ seeks to match $v_{t,\theta}$ to velocity of the neural path-interpolant $\frac{\partial\mu_{t,\eta}}{\partial t}$ and the optimal velocity $v_t^* \neq f_{t,\eta}$ since the reference process $\mathbb{Q}_\eta$ does not necessarily transport $\rho_0$ to $\rho_1$. More precisely, $\mathbb{Q}_\eta$ does not have constraints at the endpoints, that $\mathbb{Q}_0 = \rho_0$ and $\mathbb{Q}_1 = \rho_1$, which are required from our learned process $\mathbb{P}_\theta$ and its drift $v_{t,\theta}$.*

## 4 Experiments

We investigate the application of CURLY-FM on multiple applications which exhibit non-gradient field dynamics including a simple toy example, an ocean currents modeling application, a computational fluid mechanics dataset, and an application to single-cell trajectory inference. We benchmark CURLY-FM against both simulation-free flow matching approaches: Conditional flow matching (CFM) [Liu et al., 2023b, Peluchetti, 2023, Lipman et al., 2023, Albergo et al., 2023], optimal transport conditional flow matching (OT-CFM) [Tong et al., 2024a] and when possible metric flow matching [Kapuśniak et al., 2024] which cannot model non-zero drift dynamics, as well as simulation-based methods in TrajectoryNet and SBIRR [Shen et al., 2025] which can model non-zero drift dynamics but are much slower [Tong et al., 2020] and numerically unstable.

We evaluate CURLY-FM using metrics both on held out samples (2-Wasserstein ($\mathcal{W}_2$)) as well as metrics which directly measure how well the learned drift $f_\theta$ field matches the reference drift (Cosine distance and $L_2$ cost). We note that in many cases, it is not possible to match the reference drift exactly as the model is forced to match the marginals.

### 4.1 Synthetic Experiments

We start our experimental study of learning cyclical patterns from population-level observed populations by considering a synthetic example. We construct source and target distributions on asymmetrically arranged `circles` ( fig. 1a), each with higher particle population density on one side. Given a circular reference velocity field $f_t(x_t, \omega)$ with constant rotational speed, the goal is to learn the velocity-field $v_{t,\theta}(x_t)$ and trajectories $\psi_{t,\theta}(x_t)$ for $t \in [0,1]$. We find that previous flow matching methods with zero-reference field $f_t^*$ result in straight paths between source and target distributions, thereby failing to capture cycling patterns in the underlying data (see 1b and fig. 1c).

### 4.2 Modeling Ocean Currents

We model ocean currents in the Gulf of Mexico using a resolution of 1 km of bathymetry data from HYbrid Coordinate Ocean Model (HYCOM), which allows us to obtain a reference field. We follow the data processing pipeline of Shen et al. [2025] and observe 111 particles per time-point (see §D for exact dataset details). We report our quantitative results in table 2 and observe that across the left-out time points, CURLY-FM obtains the best results for the majority of the reported metrics,

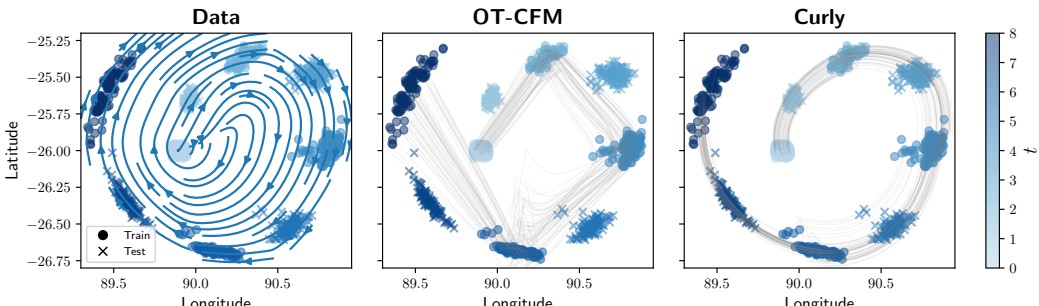

Figure 2: Visualization of ground truth data and vectorfield (left), OT-CFM predicted trajectories (center), and CURLY-FM predictions (right). Curly fits the vortex much better than OT-CFM.

Table 2: Quantitative metrics on left out test timepoints for oceans. * numbers taken from Shen et al. [2025]

| Metric | Method | $t_2$ | $t_4$ | $t_6$ | $t_8$ |
|---|---|---|---|---|---|
| EMD[1] | OT-CFM | $0.148 \pm 0.004$ | $0.227 \pm 0.008$ | $0.191 \pm 0.012$ | $0.250 \pm 0.018$ |
| | MFM | $0.107 \pm 0.014$ | $0.056 \pm 0.014$ | $0.052 \pm 0.011$ | $0.070 \pm 0.021$ |
| | Vanilla-SB* | $0.270 \pm 0.058$ | $0.300 \pm 0.056$ | $0.420 \pm 0.056$ | $0.410 \pm 0.048$ |
| | SBIRR Shen et al. [2025]* | $0.073 \pm 0.020$ | $0.072 \pm 0.012$ | $0.120 \pm 0.029$ | $0.094 \pm 0.023$ |
| | CURLY-FM | $\mathbf{0.019 \pm 0.003}$ | $\mathbf{0.045 \pm 0.005}$ | $\mathbf{0.027 \pm 0.001}$ | $\mathbf{0.030 \pm 0.006}$ |
| Cos. Dist. | OT-CFM | $0.229 \pm 0.004$ | $0.121 \pm 0.008$ | $0.034 \pm 0.005$ | $0.067 \pm 0.007$ |
| | MFM | $\mathbf{0.179 \pm 0.010}$ | $\mathbf{0.011 \pm 0.001}$ | $\mathbf{0.002 \pm 0.001}$ | $0.004 \pm 0.002$ |
| | CURLY-FM | $0.231 \pm 0.004$ | $0.017 \pm 0.001$ | $\mathbf{0.002 \pm 0.000}$ | $\mathbf{0.002 \pm 0.000}$ |
| $L_2$ cost | OT-CFM | $0.167 \pm 0.004$ | $0.144 \pm 0.014$ | $0.095 \pm 0.005$ | $0.250 \pm 0.023$ |
| | MFM | $0.203 \pm 0.011$ | $\mathbf{0.067 \pm 0.011}$ | $\mathbf{0.101 \pm 0.015}$ | $\mathbf{0.141 \pm 0.018}$ |
| | CURLY-FM | $\mathbf{0.151 \pm 0.004}$ | $0.098 \pm 0.001$ | $0.135 \pm 0.010$ | $0.178 \pm 0.017$ |

and also outperforms the previous state-of-the-art SBIRR [Shen et al., 2025] on the EMD metric. Moreover, we note that CURLY-FM is computationally fast and achieves these results in minutes compared to 4hrs for the simulation-based SBIRR. These findings are also substantiated in fig. 2, where we see trajectories that look more natural at modeling periodic behavior than OT-CFM.

## 4.3  Experiments on Single-Cell Data

To show that CURLY-FM is effective in learning dynamic behavior in single-cell data, we leverage two biologically rich datasets consisting of cell cycles in human cell fibroblasts [Riba et al., 2022] and erythroblast development in mouse [Pijuan-Sala et al., 2019]. We aim to learn cell state trajectories and development paths considering the respective RNA-velocity fields, providing information about cell cycling, lineage bifurcation, and transcriptional dynamics.

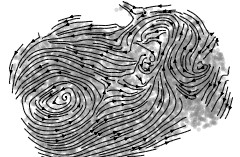 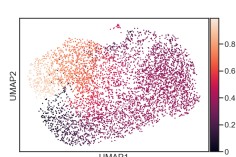

(a) RNA-Velocity Field    (b) Cell Cycles

Figure 3: Ground truth data.

**Cell cycle dynamics in human fibroblasts**. We study the nature of cell cycling in human fibroblasts and reconstruct cyclical patterns in spliced-unspliced RNA space for single genes. We leverage RNA-velocities in figure 3a to construct cell state transition paths in figure 3 by estimating RNA velocity field between marginals using $k$-nn algorithm. Further dataset details are included in §E.1. Figure 4 shows learned velocity fields $v_{t,\theta}(x_t)$ and trajectories $\psi_{t,\theta}(x_t)$ between cell cycle distributions at $t = 0$ and $t = 1$. In table 3, we show results on the trajectory inference task comparing CURLY-FM to CFM, OT-CFM, and TrajectoryNet. Given the underlying cell cycle process, the aim is to learn circular trajectories resulting from a divergence-free velocity field. While traditional methods are successful in generating end points near ground truth, they fail at learning cyclic patterns, as shown in figures 4f.

Our results show that considering a non-zero reference field and velocity inference captures non-gradient dynamics in data. Figure 4a and fig. 4d show learned behavior using CURLY-FM. We observe that the trajectory $\psi_{t,\theta}(x_t)$ inferred with CURLY-FM closely matches expected cycling patterns in the fibroblast dataset, in contrast to trajectories inferred using CFM and OT-CFM. This

is quantified in table 3, where we can see the cosine distance to the reference field is significantly lower for CURLY-FM.

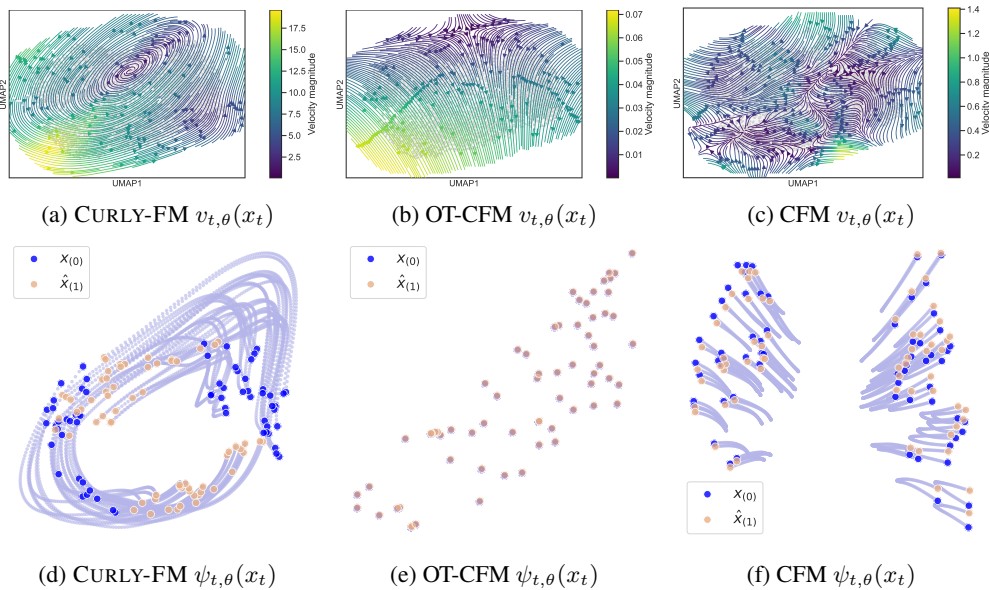

(a) CURLY-FM $v_{t,\theta}(x_t)$     (b) OT-CFM $v_{t,\theta}(x_t)$     (c) CFM $v_{t,\theta}(x_t)$

(d) CURLY-FM $\psi_{t,\theta}(x_t)$     (e) OT-CFM $\psi_{t,\theta}(x_t)$     (f) CFM $\psi_{t,\theta}(x_t)$

Figure 4: Vectorfields (top) and trajectory traces (bottom) learned using CURLY-FM (left) OT-CFM (center) CFM (right). CURLY-FM is the only method able to learn the cell cycle.

Table 3: Quantitative results for cell cycle trajectory inference task. We report the mean result for a metric with standard deviation over three seeds. CURLY-FM performs the best across matching inferred velocity field to the reference process (cosine distance) while maintaining comparable predictive quality.

| Datasets → | $d = 2$ | | $d = 10$ | | $d = 20$ | |
|---|---|---|---|---|---|---|
| Algorithm ↓ | $\mathcal{W}_2 \downarrow$ | Cos. Dist ↓ | $\mathcal{W}_2 \downarrow$ | Cos. Dist ↓ | $\mathcal{W}_2 \downarrow$ | Cos. Dist ↓ |
| CFM | $0.294 \pm 0.030$ | $1.065 \pm 0.080$ | $0.606 \pm 0.059$ | $1.001 \pm 0.037$ | $1.227 \pm 0.013$ | $1.007 \pm 0.010$ |
| OT-CFM | $\mathbf{0.248 \pm 0.030}$ | $0.800 \pm 0.309$ | $\mathbf{0.586 \pm 0.041}$ | $1.008 \pm 0.039$ | $\mathbf{1.183 \pm 0.015}$ | $0.978 \pm 0.125$ |
| TrajectoryNet | $0.531 \pm 0.021$ | $1.077 \pm 0.031$ | $0.853 \pm 0.059$ | $0.979 \pm 0.064$ | – | – |
| CURLY-FM (Ours) | $1.199 \pm 0.177$ | $\mathbf{0.295 \pm 0.040}$ | $0.930 \pm 0.024$ | $\mathbf{0.300 \pm 0.058}$ | $1.261 \pm 0.077$ | $\mathbf{0.249 \pm 0.024}$ |

**Reconstructing cell differentiation in mouse Erythroid development.** Mouse erythroid cells develop in a curved trajectory over time. We show that CURLY-FM can adhere to this developmental path purely based on velocity data for the first time. Earlier works have used manifold-based penalties to follow curved structures. We show that this is no longer necessary with clever usage of velocity information. We observe 9,815 erythroid cells undergoing differentiation and partition the data into three temporal snapshots, withholding the central marginal to assess trajectory inference (see §E.2).

We visualize trajectories in figure 5 showing that CURLY-FM clearly follows the developmental pathway of mouse erythroid cells, whereas OT-CFM fails to capture dynamics between marginals. To assess CURLY-FM performance, we measure cosine-distance and $L_2$ norm between learnt and ground truth velocities as well as $\mathcal{W}_2$ distance between points on left-out marginal. Our quantitative results show that CURLY-FM outperforms OT-CFM at reconstructing the underlying RNA-velocity field and cell trajectories in majority of selected dimensions. MFM continues to achieve

Table 4: Erythroid dataset results across dimension.

| Metric | OT-CFM | MFM | CURLY-FM (Ours) |
|---|---|---|---|
| **Dimension** $d = 2$ | | | |
| Cos. Dist | $0.146 \pm 0.001$ | $0.014 \pm 0.001$ | $\mathbf{0.009 \pm 0.000}$ |
| $L_2$ | $2.704 \pm 0.019$ | $1.999 \pm 0.014$ | $\mathbf{1.663 \pm 0.293}$ |
| $\mathcal{W}_2$ | $0.646 \pm 0.006$ | $\mathbf{0.269 \pm 0.004}$ | $0.369 \pm 0.090$ |
| **Dimension** $d = 20$ | | | |
| Cos. Dist | $0.489 \pm 0.001$ | $0.495 \pm 0.001$ | $\mathbf{0.488 \pm 0.001}$ |
| $L_2$ $(\times 10^3)$ | $1.885 \pm 0.020$ | $1.627 \pm 0.040$ | $1.721 \pm 0.035$ |
| $\mathcal{W}_2$ | $6.103 \pm 0.074$ | $\mathbf{4.855 \pm 0.052}$ | $6.124 \pm 0.027$ |
| **Dimension** $d = 50$ | | | |
| Cos. Dist | $0.490 \pm 0.000$ | $0.494 \pm 0.000$ | $\mathbf{0.489 \pm 0.000}$ |
| $L_2$ $(\times 10^3)$ | $2.215 \pm 0.022$ | $\mathbf{1.971 \pm 0.023}$ | $2.045 \pm 0.073$ |
| $\mathcal{W}_2$ | $7.969 \pm 0.029$ | $\mathbf{6.727 \pm 0.022}$ | $7.729 \pm 0.046$ |

lower $\mathcal{W}_2$, indicating stronger adherence to the underlying manifold. Conversely, CURLY-FM attains superior cosine similarity to the ground-truth velocity field, consistent with its objective emphasizing faithful velocity alignment which contributes to a challenge *exactly* matching end-point marginals.

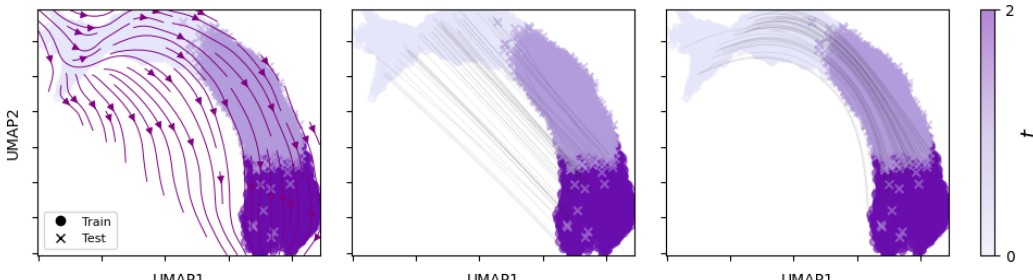

Figure 5: Visualization of ground truth data and vectorfield (left), OT-CFM predicted trajectories (center) and CURLY-FM predictions (right). Curly fits the ground truth much better than OT-CFM.

Table 5: Quantitative results for the CFD trajectory inference task. Metrics are reported on held-out particles from the test set for all marginals. Error bars show standard deviation.

| Method | Cos. Dist. $\downarrow$ | MSE $\downarrow$ | Prec.@5 $\uparrow$ | Prec.@10 $\uparrow$ | Prec.@25 $\uparrow$ |
|---|---|---|---|---|---|
| CFM | $0.254 \pm 0.003$ | $\mathbf{0.085 \pm 0.002}$ | $0.079 \pm 0.004$ | $0.164 \pm 0.006$ | $0.337 \pm 0.016$ |
| OT-CFM | $0.248 \pm 0.011$ | $0.095 \pm 0.001$ | $0.303 \pm 0.002$ | $0.388 \pm 0.004$ | $0.496 \pm 0.001$ |
| CURLY-FM | $\mathbf{0.189 \pm 0.027}$ | $0.095 \pm 0.003$ | $\mathbf{0.489 \pm 0.010}$ | $\mathbf{0.522 \pm 0.009}$ | $\mathbf{0.628 \pm 0.010}$ |

## 4.4 Experiments on Computational Fluid Mechanics Data

We evaluate CURLY-FM on a particle-based PDE dataset generated by a Lagrangian solver. Unlike grid-based (Eulerian) methods, Lagrangian approaches discretize the fluid as a set of particles that move with the flow. These particles evolve under the dynamics of the PDE which provides the particles' positions over time; other quantities of interest, such as velocity or energy, are then computed from these positions. We use data from LagrangeBench [Toshev et al., 2023], specifically the two-dimensional decaying Taylor-Green vortex (2DTGV) dataset(see §F).

In table 5, we report quantitative results on left-out particles for each marginal from a test set. We evaluate performance using (i) the cosine distance between the learned velocity field and the reference field; (ii) mean squared error (MSE) between the predicted particle positions at marginal $t+1$ and the ground truth positions, using the known coupling (ordering) between particles across marginals; and (iii) precision@$k$, measuring how often the predicted position is among the $k$ nearest neighbors of the corresponding ground truth particle. CURLY-FM outperforms baselines in terms of cosine distance and precision@$k$ while matching or outperforming the baseline methods on MSE. The smaller cosine distance for CURLY-FM shows that CURLY-FM produces velocity fields that better align with the reference field. Equal or lower MSE paired with higher precision@k shows that CURLY-FM more accurately recovers the true particle coupling and generates more faithful trajectories (c.f. fig. 10).

## 4.5 Further analysis of CURLY-FM performance

**On higher stochasticity.** We extend our discussion in section 3.1 on stochasticity levels $\sigma$ to consider an ablation where $\sigma > 0$. Despite our work being motivated in the little to no stochasticity regime, we demonstrate that considering $\sigma > 0$ does not impact CURLY-FM efficiency. We find, similar to previous work [Tong et al., 2024b], low values of $\sigma$ perform the best on all metrics. As a result, we recommend setting $\sigma$ to zero unless some reference $\sigma$ value is known. Therefore, all of our experiments are performed under $\sigma = g_t = 0$ assumption.

Table 6: Ablation on stochasticity $\sigma$.

| $\sigma$ | Cos. Dist. $\downarrow$ | $L_2 \downarrow$ | $\mathcal{W}_2 \downarrow$ |
|---|---|---|---|
| 0.01 | $0.061 \pm 0.003$ | $0.141 \pm 0.009$ | $0.028 \pm 0.066$ |
| 0.10 | $0.062 \pm 0.002$ | $0.145 \pm 0.011$ | $0.066 \pm 0.008$ |
| 1.00 | $0.145 \pm 0.009$ | $0.474 \pm 0.058$ | $0.871 \pm 0.048$ |

We consider ocean currents dataset and find that larger stochasticity monotonically decreases performance on our tasks, thus justifying our choice of $\sigma$ for our empirical work.

**On computational efficiency.** We further provide the computational cost in wall clock time for TrajectoryNet, SBIRR and CURLY-FM in table 7 (see §H.2 for further baseline comparison). We observe that CURLY-FM completes the Ocean currents problem in minutes with higher accuracy in trajectory and velocity field inference task, while SBIRR and TrajectoryNet are in the order of multiple hours and unlike CURLY-FM are simulation-based.

## 5 Related Work

**Flow matching**. Flow matching [Lipman et al., 2023], also known as rectified flows [Liu, 2022, Liu et al., 2023b] or stochastic interpolants [Albergo and Vanden-Eijnden, 2023, Albergo et al., 2023], has emerged as the default method for training continuous normalizing flow (CNF) models [Chen et al., 2018, Grathwohl et al., 2019]. However, FM can lead to unnatural dynamics less, and therefore many works attempt to derive methods for using minimum energy [Tong et al., 2024a, Pooladian et al., 2023] and more flexible conditional paths [Neklyudov et al., 2024, Kapuśniak et al., 2024].

Table 7: Compute cost.

| Method | Hours |
|---|---|
| TrajectoryNet | 7.44 |
| SBIRR | 4.67 |
| CURLY-FM (Ours) | **0.06** |

**Schrödinger bridges with deep learning**. To tackle the Schrödinger bridge problem in high dimensions many methods propose simulation-based [De Bortoli et al., 2021b, Chen et al., 2022, Koshizuka and Sato, 2023, Liu et al., 2022] and simulation-free [Shi et al., 2024, Tong et al., 2024b, Pooladian and Niles-Weed, 2023, Liu et al., 2023a] set-ups with various additional components incorporating variable growth rates [Zhang et al., 2025, Pariset et al., 2023, Sha et al., 2024], stochasticity, and manifold structure [Huguet et al., 2022] proposed based on neural ODE and neural SDE [Li et al., 2020, Kidger et al., 2021] frameworks. However, very few methods are able to incorporate approximate velocity data, and either match marginals using simulation [Tong et al., 2020], or do not attempt to match marginals [Qiu et al., 2022]. Finally, Schrödinger bridges with non-zero reference field have also been considered by Bartosh et al. [2024] and concurrently by Bartosh et al. [2025] and Shen et al. [2025], however, they do not employ a two-stage simulation-free approximation as CURLY-FM. We include further details on related work comparison in appendix §B.

**RNA-velocity methods on discrete manifolds**. A common strategy to regularize and interpret RNA-velocity [La Manno et al., 2018, Bergen et al., 2020] is to restrict it to a Markov process on a graph of cells representing a discrete manifold or compute higher-level statistics on it [Qiu et al., 2022]. However, these approaches are not equipped to match the marginal cell distribution over time. CURLY-FM can be seen as a method that unites these approaches with marginal-matching approaches.

## 6 Conclusion

In this work, we introduced CURLY-FM, a method capable of learning non-gradient field dynamics by solving a Schrödinger bridge problem with a non-zero reference process drift. In contrast to prior work, CURLY-FM is simulation-free, greatly improving numerical stability and efficiency. We showed the utility of this method in learning more accurate dynamics in a cell cycle system with known periodic behavior, computational fluid dynamics under Lagrangian solvers, and ocean currents. CURLY-FM opens up the possibility of moving beyond modeling population dynamics with simulation-free training methods and towards reconstructing the underlying governing dynamics [Xing, 2022]. Nevertheless, CURLY-FM is currently limited in its ability to discover the underlying dynamics by accurate inference of the reference field, which is an inherently difficult problem, especially over longer timescales. Exciting directions for future work involve additional verification of trajectories through lineage tracing [McKenna and Gagnon, 2019, Wagner and Klein, 2020], and improved modeling across non-stationary populations with the additional incorporation of unbalanced transport or multiomics datatypes [Baysoy et al., 2023].

## Acknowledgments

The authors acknowledge funding from UNIQUE, CIFAR, NSERC, Intel, and Samsung. The research was enabled in part by computational resources provided by the Digital Research Alliance of Canada (https://alliancecan.ca), the Province of Ontario, companies sponsoring the Vector Institute (http://vectorinstitute.ai/partners/), Mila (https://mila.quebec), and NVIDIA. KK is supported by the EPSRC CDT in Health Data Science (EP/S02428X/1). AJB and LA are partially supported by NSERC Post-doc fellowships. LA is supported by the Eric and Wendy Schmidt Center at the Broad Institute of MIT and Harvard. This research is partially supported by EP- SRC Turing AI World-Leading Research Fellowship No. EP/X040062/1 and EPSRC AI Hub on Mathematical Foundations of Intelligence: An "Erlangen Programme" for AI No. EP/Y028872/1. We thank Renato Berlinghieri, author of SBIRR, for valuable discussions and for generously sharing his ocean currents code and data, which made our ocean current experiments possible.

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

# Appendix

## A Complementary Orthogonal Work on Single Cell

There are many recent works tackling the single-cell trajectory inference problem on single-cell RNA-sequencing data. In this work we focus on incorporating a reference velocity field and learning non-gradient field dynamics. There are many other related works that may be used in conjunction with ideas in this paper. Here we detail some of these complementary works and how they could be combined with ideas in CURLY-FM. Specifically a number of areas have been identified as improving how cells are modeled by flow-based networks.

- **Optimal transport / minimal energy**: Cells are modeled by optimal transport over short enough time scales and is therefore desirable in almost all applications of single-cell trajectory inference [Hashimoto et al., 2016, Schiebinger et al., 2019, Tong et al., 2020, 2024a].

- **Density or manifold assumptions**: Cells are also known to lie on a low dimensional manifold within gene space [Moon et al., 2018]. This knowledge has been exploited in a number of works that are both require simulation during training [Tong et al., 2020, Huguet et al., 2022] and more recent methods which do not [Neklyudov et al., 2024, Kapuśniak et al., 2024]

- **Unbalanced transport (modeling cell birth and death)**: By default, flow models assume conservation of mass over time through the continuity equation. Over long time scales this is not a good model of a population of cells as [Zhang et al., 2025]

- **Stochasticity**: Cells move stochastically based on unobserved factors. This has led to a number of methods that attempt to model cells with stochastic dynamics [Koshizuka and Sato, 2023, Tong et al., 2024b, Schiebinger, 2021], and in particular with Schrödinger bridges, as the stochastic extension of dynamic optimal transport. While there have been many works on learning Schrödinger bridges simulation-free there is little work on efficient learning of general reference drift functions which we tackle here.

- **Velocity estimates**: RNA-velocity [La Manno et al., 2018] exploits particularities about the RNA collection data to measure both older and newer RNA transcripts at a single timepoint. This allows the approximation of RNA-velocity— the approximate instantaneous change of the RNA expression of each cell. First used in Tong et al. [2020], velocity estimates are relatively underutilized in the trajectory inference problem as data is more scarce and more difficult to process. To our knowledge, CURLY-FM is the first method to provide a simulation-free method for incorporating velocity information into a learned flow.

- **Distribution conditioning** Where most flow-based frameworks model cells as a non-interacting point cloud, recent work has also considered flows which include terms for the interaction between cells [Atanackovic et al., 2025, Haviv et al., 2025]. This allows the modeling of more complex interactions between cells which is an extremely prevelent dynamic in real cell systems.

While CURLY-FM focuses on the incorporation of optimal transport, stochasticity, and in particular *velocity*, it does not address problems of unbalanced transport or manifold structure of the single-cell datatype. Future work will incorporate ideas from CURLY-FM in combination with existing ideas on how to model density and unbalanced transport for more expressive, accurate, and useful models of cell dynamics towards virtual cells [Bunne et al., 2024a].

## B Further Details on Related Work

**Iterative Algorithms**. Shen et al. [2025] propose an iterative bi-level algorithm, first solving a Schrödinger bridge and estimating the forward-backward drift given a current guess of the reference drift. Given the solution to step 1, trajectories are simulated to obtain an updated estimate of the reference drift. This bi-level approach is considered since the reference drift belongs to a family of possible reference drifts rather than a single prescribed one. In contrast, CURLY-FM employs a two-stage algorithm that is not iterative and also does not search over a family of reference drifts. More critically, CURLY-FM is also simulation-free, which is achieved by making the modeling assumption that the conditional mixture of bridges is modelled as Brownian bridges. In practice, this means that

CURLY-FM is significantly more efficient to train as shown in table 15. Finally, note, both Shen et al. [2025] and CURLY-FM can model non-gradient field dynamics, but only the latter is simulation-free.

**Latent SDEs**. Bartosh et al. [2024] and Bartosh et al. [2025] consider Latent SDEs for learning stochastic dynamics with simulation-free training without explicitly solving Schrödinger Bridge problem as considered by CURLY-FM framework. This means that unlike CURLY-FM, work on Latent SDEs does not consider minimizing KL divergence to find an optimal path measure with respect to the reference process. Further, SDEs in CURLY-FM are not Latent SDEs. Whilst building a computational approach to solving the Schrödinger Bridge problem with latent SDEs may be possible, this is an orthogonal research direction to our work.

## C   Algorithmic Details

In this section we detail python code for training and inference for reproducibility.

```python
for _ in range(n_iter):
    x0 = sample_x0(batch_size)
    x1 = sample_x1(batch_size)
    x0, x1 = coupling(x0, x1)
    t = torch.rand(batch_size).type_as(x0)
    eps = torch.randn_like(x0)
    lambda_t = (2* torch.sqrt(t * (1-t))) / sigma
    xt, xt_dot = get_xt_xt_dot(t, x0, x1, geodesic_model)
    xt = xt + eps * sigma
    vt = drift_model(xt, t)
    st = score_model(xt, t)
    flow_loss = torch.mean((vt - xt_dot) ** 2)
    score_loss = torch.mean((lambda_t * st + eps) ** 2)
    loss = flow_loss + score_loss
```

Listing 1: Python implementation of CurlyFM Score and Flow Training algorithm.

```python
x = torch.randn(batch_size, dim)
for t in torch.linspace(0, 1, 100)[:-1]:
    drift = drift_model(x,t) + score_model(x,t)
    x = drift * dt + sigma * torch.sqrt(dt) * torch.randn_like(x)
```

Listing 2: Python implementation of CurlyFM inference algorithm.

## D   Ocean Currents Dataset

We assess performance of CURLY-FM on real ocean currents dataset acquired from a HYbrid Coordinate Ocean Model (HYCOM) reanalysis released by US Department of Defense.

**Dataset**. This data consists of real ocean currents measurements in Gulf of Mexico acquired at 1km bathymetry, providing hourly ocean currents velocity fields for the geographic region between 98E and 77E in longitude and 18N to 32N in latitude at each day since January 1st, 2001.

**Experimental Set-up**. We leverage experimental set-up as presented in Shen et al. [2025], and focus on a specific time point at 17:00 UTC on June 1st, 2024, extracting ocean surface velocity field that contains a vortex. From a point near its center, we uniformly draw 1,000 initial positions whithin radius 0.05 and evolve them across nine time steps such that $\Delta t = 0.9$, computing velocities by the nearest grid node for $\sim 111$ observations per time step.

**Ablations**. Tables 8 and 9 show ablations on using coupling cost in algorithm 2 for trajectory inference between drifter observations in ocean currents experiments. We observe that there is limited effect in integrating coupling cost over increased numbers of time steps between marginals.

Table 8: Comparison of CURLY-FM with and without the coupling in algorithm. 2. Without coupling refers to an independent coupling. The results are averaged over three seeds.

| Metric | Method | $t_2$ | $t_4$ | $t_6$ | $t_8$ |
|---|---|---|---|---|---|
| Cos. Dist. | CURLY-FM (without coupling) | $0.230 \pm 0.003$ | $0.017 \pm 0.001$ | $0.002 \pm 0.000$ | $0.002 \pm 0.000$ |
| | CURLY-FM | $0.231 \pm 0.004$ | $0.017 \pm 0.001$ | $0.002 \pm 0.000$ | $0.002 \pm 0.000$ |
| $L_2$ cost | CURLY-FM (without coupling) | $0.152 \pm 0.003$ | $0.099 \pm 0.001$ | $0.132 \pm 0.007$ | $0.187 \pm 0.012$ |
| | CURLY-FM | $0.151 \pm 0.004$ | $0.098 \pm 0.001$ | $0.135 \pm 0.010$ | $0.178 \pm 0.017$ |

Table 9: Comparison of different numbers of times (n) used to compute the coupling cost in algorithm 2. The times are equispaced except for $n = 1$, where they are drawn uniformly at random. The results are averaged over three seeds.

| Metric | Method | $t_2$ | $t_4$ | $t_6$ | $t_8$ |
|---|---|---|---|---|---|
| Cos. Dist. | CURLY-FM (n=1) | $0.231 \pm 0.004$ | $0.017 \pm 0.001$ | $0.002 \pm 0.000$ | $0.002 \pm 0.000$ |
| | CURLY-FM (n=3) | $0.230 \pm 0.002$ | $0.017 \pm 0.001$ | $0.002 \pm 0.000$ | $0.002 \pm 0.000$ |
| | CURLY-FM (n=5) | $0.229 \pm 0.005$ | $0.018 \pm 0.001$ | $0.002 \pm 0.000$ | $0.002 \pm 0.000$ |
| | CURLY-FM (n=10) | $0.227 \pm 0.005$ | $0.017 \pm 0.001$ | $0.002 \pm 0.000$ | $0.002 \pm 0.000$ |
| $L_2$ cost | CURLY-FM (n=1) | $0.151 \pm 0.004$ | $0.098 \pm 0.001$ | $0.135 \pm 0.010$ | $0.178 \pm 0.017$ |
| | CURLY-FM (n=3) | $0.153 \pm 0.002$ | $0.101 \pm 0.002$ | $0.132 \pm 0.004$ | $0.184 \pm 0.011$ |
| | CURLY-FM (n=5) | $0.153 \pm 0.003$ | $0.104 \pm 0.002$ | $0.131 \pm 0.001$ | $0.193 \pm 0.011$ |
| | CURLY-FM (n=10) | $0.150 \pm 0.004$ | $0.101 \pm 0.017$ | $0.130 \pm 0.007$ | $0.191 \pm 0.014$ |

## E    Single Cell Datasets

### E.1    Human Fibroblasts Dataset

We consider the human fibroblasts dataset [Riba et al., 2022] that contains genomic information about 5,367 cells observed across a fibroblast cell cycle. Cell data further contains information about cycling genes, and more specifically, their RNA velocities, which are used to estimate the reference RNA velocity field. Figure 7 shows a distribution of cell rotations and phases during a cell cycle process.

**Pre-processing**. Data is pre-processed by selecting top $d$ variable genes from the data. Further, we use `scvelo` package [Bergen et al., 2020] to compute imputed unspliced (Mu) and spliced (Ms) expressions as well as velocity graph. We construct a cell-cell $k$-nn graph using `scv.pp.neighbours(adata)` in the joint spliced and unspliced expression space using default `scvelo` hyperparameters. For each gene and cell, we compute averaged fits moment and the centered second moment of spliced and unspliced genes as well as relared attributes. RNA velocities are computed using `scv.tl.velocity(adata)` fitting the stochastic transcriptional dynamics model [La Manno et al., 2018]. Finally, we compute low-dimensional embeddings for velocities using UMAP with `scv.tl.velocity_embedding(adata)` for our visualizations in figure 4d. For selecting top $d$ highly variable genes, we use `sc.pp.highly_variable_genes(adata, n_top_genes=d)`.

### E.2    Mouse Erythroid

We consider a dataset showing mouse gastrulation subset to erythroid lineage [Pijuan-Sala et al., 2019], representing the developmental pathway during which embryonic cells diversify into lineage-specific precursors, evolving into adult organisms. The data consists of 9,815 cells evolving through five lineage stages as shown in Figure 8, and it is available through `scvelo` package API.

**Pre-processing**. Data is pre-processed using `scvelo` and `unitvelo` [Gao et al., 2022] package to compute imputed unspliced (Mu) and spliced (Ms) expressions as well as the velocity graph. With `unitvelo`, we construct the cell latent time used to approximate the differentiated time experienced by cells. Mouse erythroid velocity field is computed using `unitvelo.run_model()` following instructions from `unitvelo` documentation.

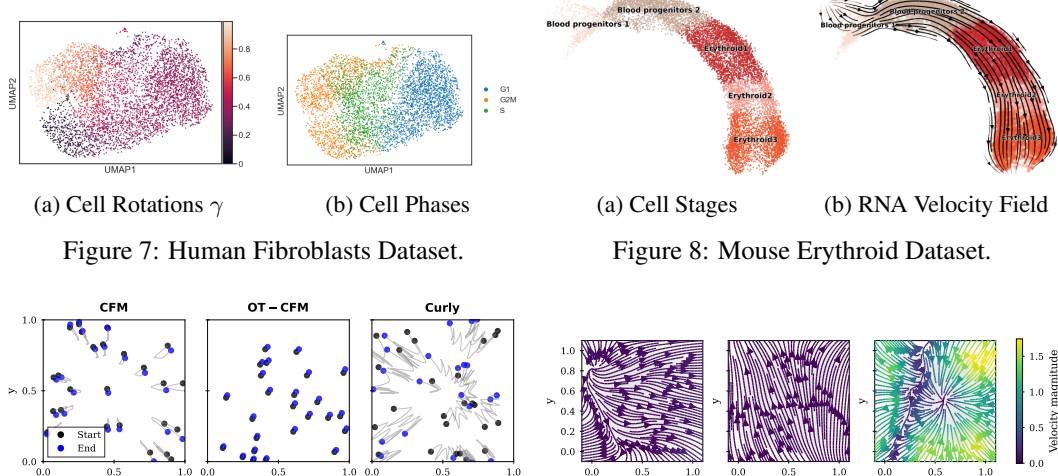

(a) Cell Rotations $\gamma$      (b) Cell Phases      (a) Cell Stages      (b) RNA Velocity Field

Figure 7: Human Fibroblasts Dataset.      Figure 8: Mouse Erythroid Dataset.

Figure 9: Trajectories of particles for CFD for different methods.

Figure 10: Learned CFD velocity field for CFM (left), OT-CFM (middle), and CURLY-FM (right).

**Filtering RNA Velocities**. To address tail-effects of noisy single-cell data, we filter RNA velocities by down-weighting distant neighbors in the $k$-NN estimate at $x_t$ and injecting small Gaussian noise. We construct new estimate $f_t^*$ by interpolating between $k$-nn velocity estimate $f_t$ and noise such that $f_t^* = (1 - w_\gamma(x_t)) * f_t + w_\gamma(x_t) * \mathcal{N}(0, 0.1)$. The weight $w_\gamma(x_t) \in [0, 1]$ is given by a sigmoid of the distance between the $k$-NN distance and a threshold hyperparameter $\gamma$, so that larger distances yield larger $w_\gamma(x_t)$ and thus stronger penalization of distant neighbors.

# F    Computational Fluid Dynamics Dataset

**Experimental details for CFD**. We conducted our CFD experiments using the two-dimensional decaying Taylor-Green vortex (2DTGV) dataset provided by LagrangeBench [Toshev et al., 2023]. We subsampled 2000 particles and considered five equispaced marginals (snapshot distributions over particle positions). The goal was to perform trajectory inference from unordered population snapshots. Since instantaneous velocity is not directly observed, we constructed the reference drift field using finite differences. This aligns with how derived physical quantities—such as velocity or energy—are computed from particle positions in Lagrangian PDE solvers.

We considered a dataset split of $[80\%, 20\%]$ across the train and test sets, respectively. For the 2000 particles, this resulted in 1600 being used for training and 400 for testing. All other hyperparameters of CURLY-FM were the same as for the other experiments.

## F.1    Ablations on CFD

Figure 9 illustrates the trajectories of 25 particles under CFM, OT-CFM, and CURLY-FM. Both CFM and OT-CFM tend to produce straight, relatively short paths, most notably in the case of OT-CFM, indicating a preference for minimal transport effort. In contrast, CURLY-FM learns longer, more intricate trajectories that better resemble the expected fluid dynamics.

Figure 10 visualizes the velocity fields inferred by each method. While CFM and OT-CFM yield smoother and simpler velocity patterns, CURLY-FM captures a richer and more structured field. This complexity reflects closer alignment with the reference field and suggests improved physical fidelity. The resulting velocity field from CURLY-FM more accurately models the underlying dynamics, adhering to both data-driven transport and the governing reference flow.

Tables 10 and 11 show CURLY-FM without coupling and for different number of times used to evaluate the coupling cost, respectively. In particular, table 10 shows that the coupling based on minimizing the kinetic energy only marginally improves performance for CFD experiments. Furthermore, table 11 shows that there is little to no benefit in using additional times to approximate the coupling cost in algorithm 2.

Table 10: Comparison of CURLY-FM with and without the coupling in algorithm. 2. Without coupling refers to an independent coupling. The results are averaged over five seeds.

| Method | Cos. Dist. ↓ | MSE ↓ |
|---|---|---|
| CURLY-FM without coupling | $0.214 \pm 0.007$ | $0.092 \pm 0.007$ |
| CURLY-FM | $0.189 \pm 0.027$ | $0.095 \pm 0.003$ |

Table 11: Comparison of different numbers of times (n) used to compute the coupling cost in algorithm 2. The times are equispaced except for $n = 1$, where they are drawn uniformly at random. The results are averaged over five seeds.

| Method | Cos. Dist. ↓ | MSE ↓ |
|---|---|---|
| CURLY-FM (n=1) | $0.189 \pm 0.027$ | $0.095 \pm 0.003$ |
| CURLY-FM (n=3) | $0.185 \pm 0.027$ | $0.092 \pm 0.003$ |
| CURLY-FM (n=5) | $0.179 \pm 0.020$ | $0.091 \pm 0.005$ |
| CURLY-FM (n=10) | $0.182 \pm 0.029$ | $0.095 \pm 0.006$ |

# G   Further Experimental Details

## G.1   Single-marginal Set-up

$\varphi_{t,\eta}(x_0, x_1, t)$ **and** $v_{t,\theta}(x_t, \eta)$ **design**. Both $\varphi_{t,\eta}(x_0, x_1, t)$ and $v_{t,\theta}(x_t, \eta)$ are designed as MLP models with 3 layers. We select MLP dimensions based on the number of chosen highly variable genes $d$. For $\varphi_{t,\eta}(x_0, x_1, t)$ we choose $d_{in} = 2 \times d$ and $d_{out} = d$. For $v_{t,\theta}(x_t, \eta)$, we choose dimensions of $d_{in} = d$. The dataset is split as $[80\%, 10\%, 10\%]$ across training, validation, and test.

**Training**. All CURLY-FM and baseline experiments are run using $lr = 10^{-4}$ learning rate and Adam optimizer with default $\beta_1, \beta_2$, and $\epsilon$ values across three seeds and with 1,000 epochs split into 500 epochs to train $\varphi_{t,\eta}$ followed by 500 epochs to train $v_{t,\theta}$.

**Baselines**. TrajectoryNet was run with 250 epochs with the Euler integrator with 20 timesteps per timepoint. We use 250 epochs to limit the experimental time and the number of function evaluations to roughly $5\times$ that of simulation-free methods. We use a batch size of 256 samples. We use a Dormand-Prince 4-5 (dopri5) adaptive step size ODE solver to sample trajectories with absolute and relative tolerances of $10^{-4}$.

**Compute**. All experiments were conducted using a mixture of CPUs and A10 GPUs.

## G.2   Multi-marginal set-up

In the multi-marginal setting, we train by randomly sampling interpolation times $t$ within intervals corresponding to each adjacent pair of marginals—for example these intervals will be [0,1] and [1,2] if our marginals lie at $t \in \{0, 1, 2\}$. For each interval $[t_i, t_{i+1}]$, we sample a random time $t$, then compute both the neural interpolant $x_t$ and its derivative $\dot{x}_t$. We also compute global time $\hat{t} = t_i + t$, if say $t = 0.5$, the global time for the second marginal pair will be $\hat{t} = 1 + 0.5 = 1.5$. This effectively ensures that neural interpolant and its derivative match the global axis of time. We then connect all the marginals by concatenating neural interpolant, its derivative and global times into single tensors and use these as inputs to train our neural path interpolant in algorithm 1 and the drift in algorithm 2.

$\varphi_{t,\eta}(x_0, x_1, t)$ **and** $v_{t,\theta}(x_t, \eta)$ **design**. We design $\varphi_{t,\eta}(x_0, x_1, t)$ and $v_{t,\theta}(x_t, \eta)$ as MLPs as shown in table 12 across all multi-marginal experiments. CFD experiments additionally use four residual connection blocks with no dropout. We select MLP dimensions based on the number of chosen highly variable genes $d$. For $\varphi_{t,\eta}(x_0, x_1, t)$ we choose $d_{in} = 2 \times d$ and $d_{out} = d$. For $v_{t,\theta}(x_t, \eta)$, we choose dimensions of $d_{in} = d$.

**Training**. We show training details for multimarginal set-up in table 12 and compute test metrics on left-out marginals. We select $k = 20$ neighbours for ocean currents and CFD experiments and $k = 30$ neighbors for mouse erythroid experiments to compute ground-truth velocities.

**Compute**. All experiments were conducted using a mixture of CPUs and A10 NVIDIA GPUs.

Table 12: Overview of model design and training hyperparameters across multimarginal experiments.

| | Mouse Erythroid | | Ocean Currents | | CFD | |
|---|---|---|---|---|---|---|
| | $\varphi_{t,\eta}(x_0,x_1,t)$ | $v_{t,\theta}(x_t)$ | $\varphi_{t,\eta}(x_0,x_1,t)$ | $v_{t,\theta}(x_t)$ | $\varphi_{t,\eta}(x_0,x_1,t)$ | $v_{t,\theta}(x_t)$ |
| Channels | 256 | 256 | 64 | 64 | 64 | 64 |
| Batch size | 256 | 256 | 64 | 64 | 256 | 256 |
| Epochs | 2k | 3k | 5k | 3k | 1.5k | 1.5k |
| Learning rate | $10^{-4}$ | $10^{-4}$ | $10^{-4}$ | $10^{-4}$ | $10^{-4}$ | $10^{-4}$ |

### G.3 Cost of OT plan

The cost of the optimal plan $\pi^*$ is intractable as it would in general require computing costs over stochastic paths. Consequently, we make several approximations to these couplings that enable faster throughput as offered in simulation-free training. In particular, we make three approximations to the cost $c(x_0, x_1)$ between two points, and one on the coupling given this cost:

- Algorithm 1 has learned a path or set of paths for the means of Gaussians in proximity to the optimal $\mathbb{Q}_t(x_t|x_0, x_1)$

- We consider low stochasticity $\sigma$ setting, and therefore the cost is close to the distance of the means travel

- The integral of the squared length of the curve is approximated empirically using a Monte Carlo estimate

After approximating $c(x_0, x_1)$, we use we use mini-batch OT to approximate the entropic OT problem following set-up in Tong et al. [2024b]. Interestingly, in the low stochasticity setting, Tong et al. [2024b] found that mini-batch OT with no stochasticity was empirically a better approximator of $\pi^*$ in the Gaussian case. This is because mini-batch transport adds some amount of "entropy" to the plan due to its approximation. We follow this approximation in our setting.

### G.4 Further details on ground-truth velocity estimate

**On use of kernels**. We highlight that in practical settings we investigate, a continuous ground truth reference field does not exist. In our applications, we only have access to the reference field at discrete points in space and time that correspond to ground truth data at the different time marginals. Consequently, we use a kernel to build a continuous reference field $f_t(x_{t,\eta})$.

**Reference drift velocity estimate** $f_t$. In our considered setting, we assume access to the ground truth reference drift at samples $x_0 \sim \mu_0$ and $x_1 \sim \mu_1$ as part of the problem setup. Note that in the high-impact application domains we consider, such as trajectory inference in single-cell data, we have access to the RNA velocity [Riba et al., 2022, Bergen et al., 2020], which is assumed to be a reasonable estimate (up to a scaling factor) of the SDE velocity [Tong et al., 2020]. As we need to estimate CURLY-FM everywhere in space and time during training, we construct a smoothed version of the reference drift by using a kernel $\kappa_t$. This allows us to construct the reference drift $f_t(\mathbf{x_t})$—using knowledge from the ground truth reference drift in the existing dataset—in places where there are no ground truth samples. Consequently, we take $f_t$ in these intermediate points as our ground truth reference drift. We further highlight that kernel estimate is not used in cases where ground-truth velocities are given on a continuous domain.

**Ablation on kernel estimate accuracy**. To show that estimate accuracy does not effect our findings, we conduct an ablation study by constructing a noisy reference drift $f_t^{\text{noise}}$ for various noise levels $\beta \in [0, 1]$. We show results in table 13 for the Ocean Currents dataset. The noisy reference drift is obtained as a linear combination of the ground truth reference drift $f_t$ and noise from a standard gaussian distribution ($f_t^{\text{noise}} = (1-\beta)*f_t + \beta*\text{noise}$). We find that the performance between $\beta = 0$ (no noise, i.e. regular CURLY-FM) and $\beta = 0.25$ are similar whilst the performance for $\beta = 0.5$ and higher gradually becomes worse, as the noise dominates over the ground truth reference drift in $f_t^{\text{noise}}$. This shows that CurlyFM is robust to moderate amounts of noise added to the ground truth reference drift.

**Reconstructed field**. We provide comparison to ground truth velocity field and $k$-nn estimate for the human fibroblasts data. From figure 12 it is clear that our approach faithfully reconstructs ground-truth RNA velocities.

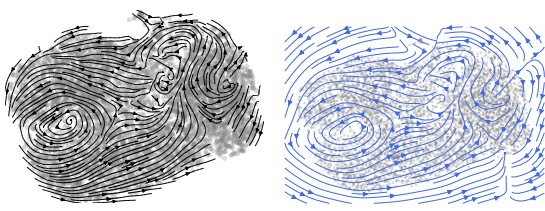

Table 13: Noisy reference drift ablation.

| $\beta$ | Cos. Dist. $\downarrow$ | $L_2 \downarrow$ | $\mathcal{W}_2 \downarrow$ |
|---|---|---|---|
| 0.00 | $0.062 \pm 0.003$ | $0.143 \pm 0.010$ | $0.034 \pm 0.006$ |
| 0.25 | $0.057 \pm 0.033$ | $0.021 \pm 0.036$ | $0.051 \pm 0.030$ |
| 0.50 | $0.087 \pm 0.047$ | $0.301 \pm 0.085$ | $0.091 \pm 0.046$ |
| 0.75 | $0.261 \pm 0.123$ | $0.381 \pm 0.120$ | $0.145 \pm 0.062$ |
| 1.00 | $0.428 \pm 0.157$ | $0.445 \pm 0.121$ | $0.237 \pm 0.079$ |

(a) Ground truth velocity      (b) $k$-nn velocity estimate

Figure 12: Ablation and estimate of ground truth velocity.

## G.5 On the tradeoff between directly computing OT plan vs. iterative refinement

Two main strategies exist for approximating the static (possibly regularized) optimal transport coupling. In this work we primarily use the mini-batch approximation. However, there are also iterative-refinement type approaches that are unbiased in the infinite limit like those presented in De Bortoli et al. [2021b], Shi et al. [2024]. These methods work by creating an "outer loop" where the bridge is simulated in one direction, then matched in the other to create an iterative refinement to match marginals. While this is possible to incorporate into our framework, it is not as suitable for our application domain, where we assume small stochasticity level. Iterative approaches are not suitable for small stochasticity levels because the number of iterations of iterative proportional fitting or iterative Markov fitting (IPF / IMF) for suitable convergence depends is inversely proportional to the stochasticity [Tong et al., 2024b, Shi et al., 2024]. Indeed, at the zero noise limit, the IMF approach collapses to a rectified flow, which only solve the OT problem in limited domains e.g. 1D [Liu et al., 2023b], and a few select Gaussian settings [Bansal et al., 2025]. For low noise levels the convergence rate is significantly slower.

## H Supplementary Results

### H.1 Additional Synthetic Experiments

**CURLY-FM robustness**. To further assess CURLY-FM robustness, we designed a toy experiment with an analytical Schrödinger Bridge solution and non-gradient dynamics based on [Tong et al., 2024b, De Bortoli et al., 2021b, Shi et al., 2024]. We bridge two Gaussians in the presence of a spiral reference field across various dimensions $d$ and stochasticity levels $\sigma = g_t = 0$. Specifically, we initialize two Gaussians in 20 dimensions centered at $\mu_0 = [-0.1, 0, 0, \ldots, 0]$ and $\mu_1 = [0.1, 0, 0, \ldots, 0]$ with standard deviations $\sigma_0 = \sigma_1 = [1, \ldots, 1]$. We also define a ground truth transport field, which unlike the standard OT field, has an additional rotational component. We note that this has equivalent marginal probability distributions, but also rotates around the origin in the second and third dimensions. This allows us to test how well our method works in a simple toy setting.

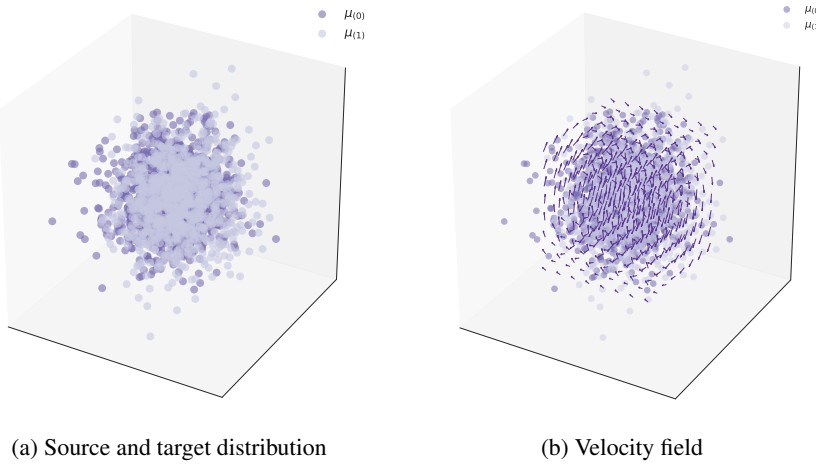

(a) Source and target distribution      (b) Velocity field

Figure 13: Example of a chosen synthetic setting in case of 3-dimensions.

Table 14: CURLY-FM ablation in synthetic setting and comparison to OT-CFM, SF2M and SB-CFM.

(a) CURLY-FM

| Dim | $\sigma$ | $KL(p_1,q_1)$ | Mean KL | Cos. Dist. | $W_2(p_1,q_1)$ |
|---|---|---|---|---|---|
| 3 | 0.0 | 0.043 | 0.024 | 0.001 | 0.192 |
| 3 | 0.1 | 0.040 | 0.017 | 0.038 | 0.610 |
| 3 | 1.0 | 0.049 | 0.049 | 0.519 | 0.613 |
| 5 | 0.0 | 0.021 | 0.017 | 0.010 | 0.871 |
| 5 | 0.1 | 0.041 | 0.018 | 0.018 | 0.880 |
| 5 | 1.0 | 0.065 | 0.038 | 0.522 | 0.887 |
| 20 | 0.0 | 0.258 | 0.178 | 0.022 | 3.887 |
| 20 | 0.1 | 0.250 | 0.169 | 0.029 | 3.891 |
| 20 | 1.0 | 0.358 | 0.234 | 0.541 | 3.749 |

(b) OT-CFM

| Dim | $\sigma$ | $KL(p_1,q_1)$ | Mean KL | Cos. Dist. | $W_2(p_1,q_1)$ |
|---|---|---|---|---|---|
| 3 | 0.0 | 0.033 | 0.024 | 0.999 | 0.369 |
| 3 | 0.1 | – | – | – | – |
| 3 | 1.0 | – | – | – | – |
| 5 | 0.0 | 0.031 | 0.021 | 1.000 | 0.879 |
| 5 | 0.1 | – | – | – | – |
| 5 | 1.0 | – | – | – | – |
| 20 | 0.0 | 0.201 | 0.165 | 0.997 | 3.921 |
| 20 | 0.1 | – | – | – | – |
| 20 | 1.0 | – | – | – | – |

(c) SF2M

| Dim | $\sigma$ | $KL(p_1,q_1)$ | Mean KL | Cos. Dist. | $W_2(p_1,q_1)$ |
|---|---|---|---|---|---|
| 3 | 0.0 | – | – | – | – |
| 3 | 0.1 | 0.017 | 0.017 | 0.480 | 0.602 |
| 3 | 1.0 | 0.024 | 0.038 | 0.509 | 0.603 |
| 5 | 0.0 | – | – | – | – |
| 5 | 0.1 | 0.038 | 0.040 | 0.516 | 1.118 |
| 5 | 1.0 | 0.108 | 0.058 | 0.518 | 1.127 |
| 20 | 0.1 | – | – | – | – |
| 20 | 0.1 | 0.537 | 0.481 | 0.515 | 4.194 |
| 20 | 1.0 | 0.572 | 0.525 | 0.510 | 4.347 |

(d) SB-CFM

| Dim | $\sigma$ | $KL(p_1,q_1)$ | Mean KL | Cos. Dist. | $W_2(p_1,q_1)$ |
|---|---|---|---|---|---|
| 3 | 0.0 | – | – | – | – |
| 3 | 0.1 | 0.029 | 0.012 | 0.999 | 0.428 |
| 3 | 1.0 | 0.028 | 0.015 | 0.998 | 0.418 |
| 5 | 0.0 | – | – | – | – |
| 5 | 0.1 | 0.026 | 0.012 | 1.000 | 0.902 |
| 5 | 1.0 | 0.024 | 0.017 | 0.997 | 0.878 |
| 20 | 0.0 | – | – | – | – |
| 20 | 0.1 | 0.233 | 0.156 | 0.995 | 3.850 |
| 20 | 1.0 | 0.264 | 0.165 | 0.998 | 3.876 |

Table 15: Computational efficiency comparison

| Method | DM-SB | Vanilla-SB | TrajectoryNet | SBIRR | CURLY-FM (ours) |
|---|---|---|---|---|---|
| Hours | 15.44 | 0.43 | 7.44 | 4.67 | **0.06** |

```python
def psi_xt(xt):
    rot_speed = 1 * np.pi
    velocity = torch.stack([
        (mu_1 - mu_0) * torch.ones_like(xt[..., 0]),
        rot_speed *  xt[...,2],
        rot_speed * -xt[...,1]
    ], dim=-1)
    if xt.shape[-1] > 3:
        extra_zeros = torch.zeros_like(xt)[..., 3:]
        velocity = torch.cat([velocity, extra_zeros], dim=-1)
    return velocity
```

Listing 3: Python implementation of ground truth vector field.

We compare against the closed-form solution and OT-CFM, measuring KL divergence, Wasserstein distance, and the cosine distance to the ideal rotational angle. Results in table 14 confirm CURLY-FM performs best in low-dimensional, low-stochasticity settings, achieving high cosine similarity.

**Comparison to baselines in synthetic setting**. We further compare CURLY-FM robustness in synthetic setting to SF2M and SB-CFM baselines. We find that CURLY-FM outperforms both SF2M and SB-CFM on stochasticity $\sigma = 0.1$ on cosine distance. In case of $\sigma = 0.1$, CURLY-FM outperforms SB-CFM, and achieves similar cosine distance with SF2M while providing better $\mathcal{W}_2$ metrics.

## H.2 Computational Efficiency

We provide comparison across DM-SB, Vanilla-SB, TrajectoryNet and SBIRR baselines in terms of computational efficiency evaluated on the ocean currents experiments (shown in table 15). We further report results for the 2D and 10D settings for TrajectoryNet and CURLY-FM in table 16. We find that TrajectoryNet is considerably slower than CURLY-FM and is not a scalable approach. This is evident in that training TrajectoryNet takes, on average, 11 times longer than CURLY-FM in the 2D case, and 17 times longer in the 10D case. Moreover, as dimensionality increases from 2D to 10D, CURLY-FM incurs an increase in computational cost by ~1.7x, whereas TrajectoryNet incurs

Table 16: Computational efficiency comparison

| Method | $d = 2$ (seconds) | $d = 10$ (seconds) |
|---|---|---|
| TrajectoryNet | $17005 \pm 110$ | $^*43080 \pm 65$ |
| CURLY-FM | $\mathbf{1429 \pm 31}$ | $\mathbf{2471} \pm 10$ |

Table 17: GSBM Control Points and Efficiency

| Method | Cos. Dist. ↓ | $L_2$ ↓ | $\mathcal{W}_2$ ↓ | Train (s) |
|---|---|---|---|---|
| GSBM 2 | $0.279 \pm 0.006$ | $0.395 \pm 0.008$ | $0.337 \pm 0.009$ | 37 |
| GSBM 10 | $0.158 \pm 0.166$ | $0.130 \pm 0.060$ | $0.247 \pm 0.144$ | 68 |
| GSBM 15 | $0.075 \pm 0.017$ | $0.134 \pm 0.030$ | $0.248 \pm 0.045$ | 68 |
| GSBM 30 | $0.088 \pm 0.034$ | $0.077 \pm 0.023$ | $0.225 \pm 0.085$ | 68 |
| GSBM 60 | $0.269 \pm 0.126$ | $0.200 \pm 0.124$ | $0.400 \pm 0.065$ | 68 |
| CURLY-FM | $0.062$ | $0.143$ | $0.034$ | 176 |

at least a 2.5x increase in computational cost. We use (*) to denote runs that did not finish within allotted resource allocation time.

## H.3 Further Baseline Comparisons

### H.3.1 Generalized Schrödinger Bridge Matching

In this section we provide a more extensive comparison of CURLY-FM to Generalized Schrödinger Bridge Matching (GSBM) [Liu et al., 2024].

**Experimental Set-up**. We make two modifications to improve GSBM in our setting for the fairest comparison to CURLY-FM: We use our loss instead of GSBM loss in equation 6a to control splines and to incorporate the signal from a reference drift directly. We fix $\sigma$ to the target $\sigma$ as we assume a constant $\sigma$ and this is the optimum for each bridge. We keep the iterative algorithm as found in the original GSBM paper. We follow the formulation in Tong et al. [2024b] which separately learns the deterministic flow and stochastic score functions, which then can be added together to calculate the stochastic drift or used separately to integrate deterministically. We note that we do this so that we can experiment with different stochasticity and simulate forward or backwards in time.

**Number of control points**. We initially set the number of control points to 2. We provide an additional ablation over the number of control points in GSBM, noting that the default number in GSBM is 15. We also find 15 provides the best tradeoff. We find that more or fewer control points do not improve performance. Otherwise we keep the same hyperparameters as CURLY-FM in terms of iterations, kernels, batch size, learning rate, models, etc. for fair comparison. Table 17 compares CURLY-FM to GSBM on the Ocean currents dataset with varying number of control points

**On loss used in baseline comparison**. We clarify that the loss (6a) in GSBM cannot take into account a non-zero reference drift directly. Loss 6a only considers potential functions $\mathbf{V}_t(x_t)$ and the kinetic energy of $u_t$, and therefore is theoretically problematic in our setting. We therefore use CURLY-FM loss function, which we believe is a more fair comparison. For completeness we further include a GSBM baseline with loss 6a found in the GSBM paper. We find that CURLY-FM is able to outperform both variations of GSBM on our oceans dataset across the three main quantitative metrics of interest. We further find our modification of GSBM's loss allows it to better match the non-zero reference drift as evidenced by a lower cosine distance, while also performing slightly better in $\mathcal{W}_1$ distance.

Table 18: GSBM loss comparison

| Method | Cos. Dist. ↓ | $L_2$ ↓ | $\mathcal{W}_2$ ↓ |
|---|---|---|---|
| GSBM (our loss) | $0.075 \pm 0.017$ | $0.134 \pm 0.030$ | $0.248 \pm 0.045$ |
| GSBM (6a) | $0.083 \pm 0.037$ | $0.134 \pm 0.029$ | $0.201 \pm 0.058$ |
| CURLY-FM | $0.070 \pm 0.001$ | $0.107 \pm 0.003$ | $0.052 \pm 0.004$ |

### H.3.2 Metric Flow Matching

For completeness of our analysis, we additionally report results for Metric Flow Matching (MFM) [Kapuśniak et al., 2024]. MFM introduces a geometric bias by enforcing interpolations that remain close to the underlying data manifold, effectively learning smooth geodesic paths that reflect intrinsic geometric structure. The two-stage learning strategy in CURLY-FM is directly adapted from MFM — replacing the manifold-constrained interpolations with regression against the reference non-gradient dynamics. In other words, CURLY-FM can be viewed as introducing an alternative inductive bias on trajectories: whereas MFM constrains paths to lie on the manifold defined by data, CURLY-FM enforces a bias on *velocities*, aiming to learn reference-consistent vector fields.

Since the core algorithmic structure of CURLY-FM (Algorithm 1) mirrors that of MFM, the two formulations are in fact compatible and can be combined—leveraging manifold-constrained interpolants together with velocity-based biases. We leave this promising direction for future work.

