# OpenReview forum: "Curly Flow Matching for Learning Non-gradient Field Dynamics"
_NeurIPS.cc/2025/Conference — NeurIPS 2025 poster_

### Official Review · Reviewer_qrFN · 2025-06-29

**Clarity:** 3
**Significance:** 3
**Originality:** 3
**Rating:** 5
**Confidence:** 4

**Summary:**

The authors proposed a generalization of flow matching method for trajectory inference without assuming a gradient field reference drift. The main idea is by decomposing Schrodinger bridge's solution into mixture of endpoint couplings and conditional bridges, and construct them in a simulation free manner separately. The authors tested their method in several different setups and seen their proposal out perform existing baselines.

**Questions:**

1) In eq (9), with general diffusion term $g_t$, should the KL divergence being $E_{P}\int_t g_t^{-1}||v||_2^2$? Similar question holds for the unnumbered equation below eq. (10).

2) I might miss understood it, as I understood the authors modeled $\mathbb{Q}$ using a mixture of Brownian bridges, with potentially different diffusion terms and $\mathbb{P}$ to have $g_t$ as diffusion term. Will the KL divergence between $\mathbb{P}$ and $\mathbb{Q}$ constructed this way being finite? KL divergence between SDEs not sharing diffusion terms might be infinite since they are not absolute continuous to each other.

3) The authors applied their method in a RNA velocity dataset treating RNA velocity as similar velocity in the SDE model. I am not sure to what extend they are connected. RNA velocity take advantage of splicing happens after transcription and use the ratio to infer whether genes are being transcribed more or less. These velocity would depends on each gene's splicing rate and mRNA's degradation rate. I wonder how they author think about these? In the ocean example one might indeed have some sparse velocity measurements but need interpolation. I encourage the authors to make some comments on how good the velocity measurements needs to be.

**Ethical Concerns:**

["NO or VERY MINOR ethics concerns only"]

**Final Justification:**

The author was able to address all my concerns and I keep my positive evaluation of this paper.

**Limitations:**

Yes mostly. But I encourage the authors to provide some comments on how velocity measurements were taken in each experiments and assumptions on them.

**Quality:**

3

**Strengths And Weaknesses:**

## Strength
1) The algorithm is simulation free and performs well.
2) Can handle none potential dynamics and leverage velocity information.

## Weakness
1) I am not fully convinced that some of the model choice is compatible with Schrodinger bridge formulation esp. when the diffusion term is different. See questions for details.
2) The notations is a bit hard to follow -- I understand this is partly due to Schrodinger bridge problems involving many different SDEs.

---

> ### Author Rebuttal · Authors · 2025-07-31
>
> We thank the reviewer for their time and effort in reviewing our work. We are pleased that the reviewer found our method to “perform well” and that the reviewer agrees that CurlyFM can handle non-gradient field dynamics by leveraging velocity information. We now address the main questions posed in the review.
>
>
> ## Compatibility of the model with Schrödinger Bridge
>
> We acknowledge the reviewer's questions regarding the compatibility of our diffusion terms $g_t$ in various places of our Schrödinger Bridge formulation. We now answer each question in turn.
>
> > In eq (9), with general diffusion term , should the KL divergence being $E_P \int_t g_t^{-1} \|v\|^2_2$? Similar question holds for the unnumbered equation below eq. (10).
>
> The reviewer is indeed correct in their observation. However, for our problem, we can drop the scaling factor as it doesn’t affect the optimization problem, so we drop this term for simplicity. We will add this as a note to clarify this point..
>
> > Compatibility with different diffusion terms
>
> The reviewer is again correct in their statement that SDEs in the SB problem that do not share diffusion terms could lead to problems. However, in this work, we consider the KL divergence between SDEs with matching diffusion terms, and always construct $\mathbb{Q}$ as a mixture of Brownian bridges with the same diffusion term as $\mathbb{P}$. We hope that this clarifies this technical point.
>
> ## Velocity data in transcriptomics and oceans datasets
>
> We thank the reviewer for their helpful suggestions here. We are happy to provide further discussion on velocity measurements and the assumptions on them in each application. The reviewer is correct to point out that the assumptions on velocity measurements may be quite different between applications, and the quality of the velocities may vary greatly as well.
>
> ### Relating RNA velocity to velocity in the SDE model
>
> We agree with the reviewer that RNA velocity attempts to compute the rate of transcription of each gene, which is not directly analogous to the velocity used in the SDE model. We follow [1] in assuming that the RNA velocity estimate is a reasonable estimate (up to a scaling factor) of the SDE velocity. There are a number of discrepancies between these two types of velocities, but they nevertheless seem correlated enough in practice to provide improved estimates of the SDE velocity field as demonstrated in a number of subsequent works [2, 3, 4].
>
> Intuitively, this interpretation makes sense, as assuming we have a perfect estimate of transcription and degradation rates and at an infinite number of locations, then following the continuous vector field in RNA state space defined by these rates should be a reasonable model of cell dynamics (at least at the RNA level). These, of course, are unrealistic assumptions, but they give intuition on how RNA velocity estimates may be useful in SDE models of cell dynamics.
>
> ### Sparse velocity interpolation
>
> Yes, we thank the reviewer for their insightful comment on velocity interpolation. This is indeed a necessary step in both the oceans and single-cell experimental setups—the most realistic setting—we consider in CurlyFM, as most velocities will be measured at sparse locations both in time and space.
>
> In this work, we primarily explore how to use a continuous velocity prior in flow matching models and leave detailed exploration of this interpolation to future work. More precisely, we rely on using simple kernel smoothing following [5] for all applications with sparse velocities.
>
> ### How good do the velocity measurements need to be?
>
> We thank the reviewer for their insightful remark on how the quality of velocity measurements might affect the performance of our model. To investigate this, we ablate the velocity quality by adding various levels of noise. In particular, we construct a noisy drift  $f_{t}^{\text{noise}}$ as a linear combination of the ground truth reference drift $f_{t}$ and noise from a standard gaussian distribution ($ f_{t}^{\text{noise}} = (1 - \beta) * f_{t} + \beta * \text{noise}$). Note that $\beta = 0$ (no noise) corresponds to the setting of CurlyFM used in the main paper. The results of this ablation are presented in the Table R1.
>
> We find that the performance between $\beta = 0$ (no noise, i.e. regular CurlyFM) and $\beta = 0.25$ (25\% noise) are similar whilst the performance for $\beta = 0.5$ and higher gradually becomes worse, as the noise dominates over the ground truth reference drift in $f_{t}^{\text{noise}}$. This shows that CurlyFM is robust to moderate amounts of noise added to the ground truth reference drift.
>
> Table R1: Ablation with noisy reference drifts
> | $\beta$ | Cos. Dist.     | L2 cost       | W1            |
> |--------:|----------------|---------------|---------------|
> | 0.00    | 0.062 ± 0.003  | 0.143 ± 0.010 | 0.034 ± 0.006 |
> | 0.25    | 0.057 ± 0.033  | 0.021 ± 0.036 | 0.051 ± 0.030 |
> | 0.50    | 0.087 ± 0.047  | 0.301 ± 0.085 | 0.091 ± 0.046 |
> | 0.75    | 0.261 ± 0.123  | 0.381 ± 0.120 | 0.145 ± 0.062 |
> | 1.00    | 0.428 ± 0.157  | 0.445 ± 0.121 | 0.237 ± 0.079 |
>
>
> ## Concluding remarks
>
> We would like to thank the reviewer again for their time reviewing our paper. We hope that our rebuttal here was successful in answering the reviewer's great points and allows the reviewer to continue positively endorsing our paper. We are also more than happy to answer any further questions that arise in the discussion phase.
>
> ## References
>
> [1] Tong, Alexander, et al. "Trajectorynet: A dynamic optimal transport network for modeling cellular dynamics." International conference on machine learning. PMLR, 2020.
>
> [2] Koshizuka, Takeshi, and Issei Sato. "Neural lagrangian schr\" odinger bridge: Diffusion modeling for population dynamics." arXiv preprint arXiv:2204.04853 (2022).
>
> [3] Neklyudov, Kirill, et al. "A computational framework for solving wasserstein lagrangian flows." arXiv preprint arXiv:2310.10649 (2023).
>
> [4] Qiu, Xiaojie, et al. "Mapping transcriptomic vector fields of single cells." Cell 185.4 (2022): 690-711.
>
> [5] Shen, Yunyi, Renato Berlinghieri, and Tamara Broderick. "Multi-marginal schr\" odinger bridges with iterative reference refinement." AISTATS (2025).

---

> > ### Comment · Reviewer_qrFN · 2025-08-02
> >
> > I appreciate the authors' detailed responses that address vast majority of my questions. I encourage the authors to include some of the discussions esp. assumptions on velocity into a revision. Again, my assessment towards this work remains positive -- the authors did a good job.
> >
> > One last follow up question on the remark of dropping $g_t$ in optimization. I understand when $g_t$ is a simple scalar one can take it out from the integral and expectation and won't influence the KL optimization target. But this is not the case when $g_t$ depends on time $t$ (and potentially state $X$), it can no longer be taken out from the integral. An extreme example is that $g_t = 1$ when $t\in [0, 0.5)$ and $g_t=10^{40}$ when $t\in [0.5, 1]$, the second half of the time is essentially ignored since it has very low weight $g_t^{-1}$ but dropping $g_t$ effectively set it to be all $1$ and the weights are the same for both part of the interval.

---

> > > ### Author Response · Authors · 2025-08-02
> > > **Response to reviewer response**
> > >
> > > We thank the reviewer for reading our rebuttal and their continued positive endorsement of our paper—we greatly appreciate it! We now answer the few remaining minor points raised by the viewer.
> > >
> > >
> > > ## Dropping $g_t$ in optimization
> > >
> > > We fully agree with the reviewer that in cases where $g_t$ depends on time $t$ and also potentially the state $\mathbf{X}$, we cannot take it out of the integral. This is a modeling assumption in our setup driven by both the fidelity and scalability of empirical performance. Consequently, our handling of $g_t$ is similar in spirit to loss weighting in models like DDPM [1] which set $\lambda = 1$ for simplified loss calculations leading to higher performance generative modeling of natural images rather than the appropriate ones that lead to the tightest ELBO.
> > >
> > > We thank the reviewer again for their comments and we will include a larger discussion on this point in the updated draft of the paper.
> > >
> > > ## References
> > >
> > > [1] Ho, Jonathan, Ajay Jain, and Pieter Abbeel. "Denoising diffusion probabilistic models." NeurIPS. 2020

---

> > > > ### Comment · Reviewer_qrFN · 2025-08-07
> > > >
> > > > Thank the authors for this response. I understand this a reasonable choice can be made. My main request would be to be clear that such a choice was made at some point as an approximation and it performs well which is already promised. As my score is already positive, I will maintain it.

---

### Official Review · Reviewer_9hEa · 2025-07-01

**Clarity:** 1
**Significance:** 2
**Originality:** 2
**Rating:** 4
**Confidence:** 4

**Summary:**

In this paper, the authors propose a numerical methodology to solve the standard dynamic Schrödinger Bridge (SB) problem ($\min KL(P|Q):P_0=\pi_0, P_1=\pi_1$), in the particular case where the reference process $Q$ has non-zero (or at least non-linear) drift. This is motivated by the fact that various real-world dynamics exhibit non gradient, periodic behavior, which cannot be modeled by simple reference dynamics. In theory, the solution to this problem writes as the mixture of the induced entropic OT plan and the reference "bridge" process $Q_{|0,1}$ (i.e., the reference process pinned at terminal times 0 and 1). The goal of the method is to approximate the drift of this solution by a neural network to be able to generate dynamics afterwards. A popular and simulation-free approach to compute the SB problem is bridge matching, as proposed in [1], which relies on two requirements wrt $Q$ : (i) being able to sample from the conditional distribution $Q_{t|0,1}$ at any time $t$ and (ii) having a tractable expression of the drift of $Q_{|0,1}$ (which is Markovian). In the current setting, these two requirements are not satisfied.

To circumvent this issue, the authors propose to approximate the reference bridge by a neural interpolant (i.e., an interpolant that relies on a neural network) before applying the standard bridge matching technique, in a similar way to [2]. This ends up in a two stage procedure coined "Curly Flow Matching", which requires two neural networks (one to approximate the reference bridge, one to approximate the drift of the SB solution). Then, they present the results of numerical experiments on various types of dynamics (single-cell trajectory, fluid dynamics...) and compare their method to classic flow matching and mini-batch OT-based flow matching.

[1] Diffusion Schrödinger Bridge Matching, Shi et al., Neurips 2024.

[2] Generalized Schrödinger Bridge Matching, Liu et al. ICLR 2024.

**Questions:**

**Questions**
- from what I understand, the first stage of Curly-FM consists in approximating the marginal distribution of the reference bridge (i.e. $\mathbb{Q}_{t|0,1}$) by a deterministic neural interpolant. Since $\sigma_t$ (the volatility of the reference process) is not zero, I have the feeling that there is a mismatch between what needs to be approximated (a probability distribution) and the approximation itself (a Dirac mass). In this case, the resulting SB solution is not correct, since it is not stochastic. Do I miss something ? In my opinion, a more adapted parameterization of the marginal distribution of the reference bridge would be a Gaussian distribution whose mean is given by the proposed deterministic interpolant with parameterized variance. Of course, this would change the matching losses (since the drift of the corresponding bridge process would not simply given by the time derivate of the Gaussian mean).
- I do not understand how the loss of Alg. 1 is built : putting aside my previous point (ie considering the deterministic case only), the objective function $\mathbb{E}[\|\partial_t X_t,\eta-f_t (X_{t, \eta})\|^2]$, where the expectation is taken wrt $(X_0, X_1, X_{t, \eta})$ and $f_t$ is exactly the drift of the reference process, is perfectly legitimate in my opinion to approximate the reference bridge. In the stochastic case, this would correspond to minimize wrt $\eta$ the objective $KL(\mathbb{Q^\eta}|\mathbb{Q})$, where $\mathbb{Q^\eta}=\int \mathbb{Q^\eta}(\cdot|x_0, x_1)d\pi(x_0,x_1)$,  as also proposed in [1] for the general SB setting. Why justifies the use of kernels here ?
- I do not understand how the cost of the OT plan can be deduced. Regarding the formulation of the SB problem, the optimal plan $\pi^\star$ is given by $\min KL(\pi|Q_{0,1})$ (with the right marginals), where $Q_{0,1}$ is the joint distribution of the reference process at times 0 and 1 (which is not tractable). Even when using the approximated bridge (defined via the neural interpolant), this joint distribution remains untractable and an additional entropy term is missing in the loss. How do you obtain this OT cost ? How do you explain that you obtain a non-regularized OT cost, while the SB problem naturally leads to an entropy-regularized OT problem when reducing to the couple (0,1) ?
- do you consider different values for $\sigma_t$ and $g_t$ ? How these values are set in the experiments ? How does the algorithm behave when changing these values ?
- the tradeoff between directly computing the OT plan (for example with Sinkhorn algorithm or mini-batch OT) + applying bridge matching VS doing an iterative refinement of bridge matching loss, by taking the current outputs of the model (see [1, 2]) has not been obviously solved in the past years: have you tried to turn Curly-FM into an iterative algorithm ?


**Suggestions**
- I would suggest the authors to write "non-linear drift" rather than "non-zero drift", which is more accurate since the DSBM approach also provides expressions for the first case, which is more general (see Appendix B.2)
- I would recommend the authors to put more highlight on the mathematical formulation they want to solve, in particular on the choice of the volatilities $\sigma_t$ and $g_t$ : is it any positive couple ? is the same value for both volatilities ? do you always consider the zero-limit ?
- I would also suggest the authors to rewrite Section 3.2 in a clearer manner (for example, explicitly define two parts : (i) learning the bridge and (ii) learning the drift of the SB solution) to include the computation steps that justify the way the algorithm is built

[1] Generalized Schrödinger Bridge Matching, Liu et al. ICLR 2024.

[2] Schrödinger Bridge Flow for Unpaired Data Translation. De Bortoli et al, 2024

**Update**: these questions have been addressed by the authors in the discussion and imply changes in the manuscript.

**Ethical Concerns:**

["NO or VERY MINOR ethics concerns only"]

**Final Justification:**

The authors have addressed most of my concerns (described in my final answer in the discussion). I increase my score based on the integration of the elements given by the authors in the manuscript.

**Limitations:**

I think that the paper lacks detailed ablation studies on the proposed algorithm to evaluate its robustness (toy setting with increasing dimension, various values of volatility...). For example, it would be interesting to have an experiment, where the reference bridge can be analytically computed (i.e linear reference SDE) and compared with the output of Algorithm 1. This would enable to have some empirical guarantee on the first stage of Curly-FM.

**Update**: these questions have been addressed by the authors in the discussion and imply changes in the manuscript.

**Paper Formatting Concerns:**

No concerns.

**Quality:**

2

**Strengths And Weaknesses:**

**Strengths**:
- the motivation of tackling the SB problem with non-zero drift is well explained
- the range of the numerical experiments is interesting and based on real-world systems

**Weaknesses**:
- in my opinion, the mathematical explanation of the Curly-FM algorithm (Section 3.2) lacks crucial details on the manner the losses are defined (I could not find any additional details in the appendix), which makes the reading very hard. While the authors claim to solve the SB problem (which is stochastic by nature, due to the consideration of non-zero volatilities), the proposed method seems to only solve the deterministic-limit case (i.e. when the volatilities go to zero)... In particular, I have serious doubts about (i) the loss to approximate the reference bridge and (ii) the way the OT plan is approximated (see section "Questions") to solve an arbitrary SB problem with non-linear drift.
-  in [1], the authors propose a very close approach to Curly-FM in a general SB setting, where the reference process does not have a trivial drift too (i.e., (1) approximation of the reference bridge and (2) application of bridge matching). The key advantage in comparison with the current method is that the reference bridge is approximated by a Gaussian Stochastic interpolant (which has since been extended to interpolants with Gaussian mixtures) which does not rely on any neural parameterization (rather a spline parameterization). This method should be compared with in the numerical experiments, especially via the trade-off expressivity/computational budget.

[1] Generalized Schrödinger Bridge Matching, Liu et al. ICLR 2024.

---

> ### Author Rebuttal · Authors · 2025-07-31
>
> We thank the reviewer for their time and effort in reviewing our work. We are glad the reviewer found our work “well explained” and found the range of experiments interesting. We now address the main concerns raised in the review, grouped by theme.
>
> ## Motivation for CurlyFM
>
> Our work is motivated by practical applications, which are similar to Schrödinger bridge problems and motivated by them, but have different criteria for success. Namely, we aim to model dynamical systems well empirically in a variety of diverse settings, where the reference drift may only be observed at measurements rather than everywhere in space and in time. This necessitates the need to devise a learning framework that is efficient to scale to these problems and to iterate quickly, two points that we have sought to demonstrate empirically through our selection of benchmarks and results.
>
> In more detail, our work attempts to find a computationally efficient solution to model our high-impact application settings, such as trajectory inference in single-cell RNA with RNA velocity, rather than computing the best numerical approximation to the Schrödinger bridge problem in a general setting. Given the vast and storied literature on SBs with many numerical approximations that require simulations to solve, we position CurlyFM as a simulation-free method that takes key ideas from the SB problem and allows modeling of complex non-gradient field dynamics found in real-world systems, which may have relatively low noise relative to the scale of the drift.
>
> Also of note, in these applications, previous work [1] has found low or no stochasticity to work the best empirically. Consequently, most of our approximations are optimized assuming that we will likely be in the low or no-stochasticity settings, as motivated by our choice of experimental benchmarks. We understand that this motivation may not have been clear in our original draft, and we will update the paper accordingly.
>
> ## Approximation of SB problem in the stochastic case
>
> We thank the reviewer for their insightful question and opportunity to clarify our work. Despite our work being motivated in the little to no stochasticity regime, we now consider an ablation below where $\sigma > 0$ to demonstrate that this is a simple modification to CurlyFM that does not lose any of its efficiency.
>
> ### Behavior with respect to $\sigma_t$ and $g_t$
>
> We find, similar to previous work [1], that low values of $\sigma$ perform the best on all metrics. As a result, we recommend setting $\sigma_t$ to zero unless some reference $\sigma$ value is known. Therefore, all of our experiments in the current manuscript are with $\sigma_t=g_t=0$. To show this in our setting, we perform a new ablation of $\sigma_t$ and $g_t$ on the oceans application. We find that larger sigma monotonically decreases performance on our tasks, justifying our choice of $\sigma_t=0$ for our empirical work.
>
> Table R1: Ablation on stochasticity for CurlyFM
> |$\sigma$|Cos. Dist.|L2 cost|W1|
> |-|-|-|-|
> |0.01|0.061 ± 0.003|0.141 ± 0.009|0.028 ± 0.066|
> |0.10|0.062 ± 0.002|0.145 ± 0.011|0.066 ± 0.008|
> |1.00|0.145 ± 0.009|0.474 ± 0.058|0.871 ± 0.048|
>
> We further note that bridge matching approaches based on iterative refinement do not converge well with small values of $\sigma$ and therefore are only really practical in the large $\sigma$ setting [1]. In settings where a large $\sigma_t$ is required, we believe that these methods may be more well-suited to the task, but present a different set of tradeoffs with better theoretical approximation of the bridge, but in contrast, may incur larger computational complexity due to simulation.
>
> ## Comparison with GSBM
> We appreciate the reviewer’s suggestion on the inclusion of GSBM as a baseline. We now include this baseline adapted to the Ocean currents dataset below using spline neural interpolant $x_t$ (as in GSBM) as well as a comparison of the training times between GSBM and CurlyFM.
>
> Table R2: Comparison with GSBM
> | | Cos. Dist. | L2 cost | W1 | Train time (s) |
> |-|-|-|-|-|
> | **GSBM** | 0.279 ± 0.006 | 0.395 ± 0.008 | 0.337 ± 0.009 | **37** |
> | **CurlyFM**| **0.062 ± 0.003** | **0.143 ± 0.010** | **0.034 ± 0.006** | 176 |
>
> We see that CurlyFM performs significantly better than GSBM, albeit at a higher cost which represents the expected tradeoff between expressivity and performance. We will include these baselines in our update to the paper.
>
> ## Use of kernels
> The reviewer is correct that if we have direct access to the reference $f_t(x)$ which is continuous in time and space, then kernels are unnecessary. However, in the practical settings that we investigate, a continuous ground truth reference field does not exist $f_t(x)$. In our applications, we only have access to the reference field at discrete points in space and time that correspond to ground truth data at the different time marginals. Consequently, we use a kernel to build a continuous reference field $f_t(x)$. We will clarify this further in the main text and thank the reviewer for their comments on how to improve the readability of our work, and we will add motivation for each step in the methods section.
>
> ## Cost of OT plan
> The reviewer makes an insightful observation that the optimal plan $\pi^* = \min KL( \pi | Q_{0,1})$ is indeed intractable. Consequently, we make several approximations to these couplings that enable faster throughput as offered in simulation-free training. In particular, we make three approximations to the cost $c(x_0, x_1)$ between two points, and one on the coupling given this cost. These are the following:
>
> - 1) Algorithm 1 has learned a path or set of paths for the means of Gaussians not too far from the optimal $Q_t(x_t | x_0, x_1)$.
> - 2) The diffusion term $\sigma_t$ is small, and therefore the cost is close to the distance the means travel.
> - 3) The integral of the squared length of the curve is approximated empirically using a Monte Carlo estimate
>
> After approximating $c(x_0, x_1)$, we use mini-batch OT to approximate the entropic OT problem following [1]
>
> Interestingly, in the low stochasticity setting [1] found that mini-batch OT with no stochasticity was empirically a better approximator of $\pi^*$ in the Gaussian case. This is because mini-batch transport adds some amount of “entropy” to the plan due to its approximation. We follow this approximation in our setting. We hope that this clarifies some of the approximations made in the OT computation for our method, and will add a detailed explanation of these choices to the appendix, as well as a brief explanation in the main text.
>
> ## Tradeoff between directly computing OT plan vs. iterative refinement
> This is an interesting question. We have not tried turning CurlyFM into an iterative refinement-based algorithm. However, we argue that this is a fairly easy adaptation, but not suitable for our application domains, where we assume small stochasticity levels. We also note that iterative refinement-type algorithms all require extensive simulation for each refinement step, and hence are not really simulation-free methods, which is a key goal here. However, we do acknowledge that iterative refinement-based approaches can still be significantly faster than methods that require backpropagation through simulation (e.g. [2,3]). We will update this within our related work and also discuss the tradeoffs depending on stochasticity levels in the appendix.
>
> ## Other suggestions around clarifications
> We appreciate the reviewer's thoughtful suggestions, and we will update the manuscript accordingly, taking into account all three suggestions.
>
> ## Ablation to evaluate its robustness in a toy setting
> We thank the reviewer for their insightful suggestion. To assess robustness, we designed a toy experiment with an analytical SB solution and non-gradient dynamics based on [1,4,5]. We bridge two Gaussians in the presence of a spiral reference field across various dimensions $d$ and noise levels $\sigma_t​=g_t$​. We compare against the closed-form solution and OT-CFM, measuring KL divergence, Wasserstein distance, and the cosine distance to the ideal rotational angle. Results confirm CurlyFM performs best in low-dimensional, low-stochasticity settings, consistently achieving high cosine similarity.
>
> Table R3: CurlyFM Ablation
> |Dim|$\sigma$|KL$(p_1, q_1)$|Mean KL|Cos. Dist.|$W_2(p_1, q_1)$|
> |-:|-:|-:|-:|-:|-:|
> |3|0|0.043|0.024|0.001|0.192|
> |3|0.1|0.04|0.017|0.038|0.61|
> |3|1|0.049|0.049|0.519|0.613|
> |5|0|0.021|0.017|0.01|0.871|
> |5|0.1|0.041|0.018|0.018|0.88|
> |5|1|0.065|0.038|0.522|0.877|
> |20|0|0.258|0.178|0.022|3.887|
> |20|0.1|0.25|0.169|0.029|3.891|
> |20|1|0.358|0.234|0.541|3.749|
> |OT-CFM Baseline|
> |3|0|0.033|0.024|0.999|0.369|
> |5|0|0.031|0.021|1|0.879|
> |20|0|0.201|0.165|0.997|3.921|
>
> ## Concluding remarks
> We thank the reviewer again for their time, effort, and thoughtful suggestions, which have enabled us to clarify technical details and also include ablations and baselines that strengthen the empirical results. We further hope that this rebuttal was successful in answering all the reviewer's great points and alleviating the concerns shared by the reviewer. If the reviewer is satisfied, we invite the reviewer to consider a fresher evaluation of our work with this rebuttal in context. We are also more than happy to answer any further questions the reviewer might have. Please do let us know.
>
> ## References
> [1] Tong, Alexander, et al. "Simulation-free schrödinger bridges via score and flow matching." AISTATS. 2024.
>
> [2] Tong, Alexander, et al. "Trajectorynet: A dynamic optimal transport network for modeling cellular dynamics." ICML. 2020.
>
> [3] Koshizuka, Takeshi, and Issei Sato. "Neural lagrangian schrödinger bridge: Diffusion modeling for population dynamics." ICLR. 2023.
>
> [4] De Bortoli, Valentin, et al. Diffusion Schrödinger bridge with applications to score-based generative modeling. NeurIPS. 2021
>
> [5] Shi, Yuyang et al. Diffusion Schrödinger bridge matching. NeurIPS. 2023

---

> > ### Comment · Reviewer_9hEa · 2025-08-04
> > **Answer to the rebuttal**
> >
> > Thank you for addressing my concerns, I will respond to the rebuttal item per item.
> >
> > 1. **About the deterministic variant of the GSB problem**. Given the motivation formulated by the authors, I now understand that the proposed method is meant to solve the *deterministic* limit of the Schrödinger Bridge problem with non-linear drift, rather than the general stochastic problem. I think this should be much more highlighted in the introduction and in the framework section to avoid any misleading since it completely guides the algorithmic pieces. It should be clearly indicated that the volatility coefficients are equal to 0 in the experiments (it was not mentioned in the submission), as indicated by the authors.
> >
> > From what I understand, the authors however still consider the possibility of small stochasticity for their algorithm (since they provide ablation studies on $\sigma_t$ and additional experiments with volatility); but, at the same time, they do not address my first question on the design of the bridge in this case, which is expected to be stochastic but is actually approximated by a deterministic function. I will reiterate it below :
> > > from what I understand, the first stage of Curly-FM consists in approximating the marginal distribution of the reference bridge (i.e. $\mathbb{Q}_{|0,1}$) by a deterministic neural interpolant. Since (the volatility of the reference process) is not zero, I have the feeling that there is a mismatch between what needs to be approximated (a probability distribution) and the approximation itself (a Dirac mass). In this case, the resulting SB solution is not correct, since it is not stochastic. Do I miss something ?
> >
> > As far as I understand, this mismatch may be an explanation for the poor performance of the stochastic variants in the provided ablation study.
> >
> > 2. **About the comparison with GSBM.** In its original version, GSBM is an iterative algorithm that alternates between learning the bridge and learning the velocity field. Do you keep this instantiation in your comparison ? How many control points do you use ? It would be interesting to have the detail of the hyperparameters to ensure a fair comparison.
> >
> > 3. **About the use of kernels.** Thank you for the clarification. As you indicate, this should be definitely well highlighted in the manuscript.
> >
> > 4. **Approximation of the OT cost.** Thank you for the clarification, it make more sense now. As I specified in my review, I think this is really important that the authors take the time to detail their assumptions and approximations (either in the main or the appendix)
> >
> > 5. **About the iterative adaptation of CurlyFM.** I agree with the authors that the extension to an iterative approach is pretty straightforward, but is not necessarily needed as such in the current paper. However, I do not understand the reason invoked by the authors to not perform it :
> > > it is not suitable for our application domains, where we assume small stochasticity levels
> >
> > Indeed, with zero-stochasticity, this would be like a Reflow procedure (ie finding the non-regularized OT plan by refinement of Flow matching). Do you see an issue there ?
> >
> > 6. **About the additional experiment.** Thank you for following my suggestion. Could you also add the baselines SB-CFM and SF^2M for the stochastic regime to bring full consistency and to see if CurlyFM does improve on them ?

---

> ### Comment · Reviewer_9hEa · 2025-08-07
> **Answer to Comment Part 1/2**
>
> Thank you for pursuing the discussion. I will try to answer point by point.
>
> 1. **About the deterministic variant of the GSB problem.** Thank you for detailing the set up of the stochastic (and canonical) version of the bridge in the SB problem. To be clear, my concern was related to the fact that only a deterministic setting of the bridge was given (see Equation 11 in the main), and no extension to the stochastic setting was given besides this in the paper. As I understand, when the volatility of the reference dynamics $g_t$ is positive, the authors propose to model the bridge as $Q_{t|0,1}= N( x_t; tx_1 + (1 - t)x_0 + t(1 - t)φ{t, \eta}(x_0, x_1), g^2_t)$. However, I am afraid that this modeling is not correct and surely introduces numerical bias when the volatility increases. For instance, think of the case where the reference is zero and $g_t=\sigma$, where $\sigma>0$ is constant: in this case, the bridge $Q_{t|0,1}$ is tractable (this is a scaled Brownian bridge), and its marginal variance is given by $\sigma^2 t(1-t)\neq \sigma^2$. Hence, even in a simple case, the proposed modeling is not able to provide a good approximation when $g_t$ is not very low. For general bridge, the marginal variance should notably cancel out at the boundaries, which is not the case with the proposed modelling. As I explained in my first review, a well-suited formulation would be to follow the setting of GSBM, which lets flexible the variance and learns it along the marginal mean. I have the feeling that this set up explains why the proposed approach cannot work with non-zero volatility.
>
> 2. **About the comparison with GSBM.** Following my first point, I am afraid that the comparison you propose with GSBM is not fair, especially related to the fact that you use the loss at Line 162, instead of their variational loss (6a) which is more theoretically grounded. Indeed, when the bridge interpolant $x_{t,\eta}$ has a variance (which is the case that we treat here), then the exact drift of the corresponding marginal-preserving SDE is not simply the conditional expectation of the interpolant given by $\mathbb{E}[x_{t,\eta}|x_0,x_1]$ -> **this quantity is the drift of the corresponding marginal-preserving ODE** (which amounts to the deterministic case advocated by the authors). To compute the drift of the SDE, one has to take into account an additional term in the conditional expectation, which is related to the variance of  $x_{t,\eta}$ -> see Equation (8) in GSBM paper. Hence, this implies a modification of the loss $\mathcal{L}(\eta)$ proposed by the authors, which , as I explained in my  original review, is only valid in the deterministic case.

---

> > ### Author Response · Authors · 2025-08-07
> >
> > We thank the reviewer for continuing the discussion and allowing us to clear up what we believe are few remaining misunderstandings.
> >
> > > As I understand, when the volatility of the reference dynamics $g_t$ is positive the authors propose to model the bridge as
> > $Q_{t|0,1} = N(x_t; t x_1 + (1 -t) x_0 + t (1 - t) \psi t,\eta(x_0, x_1), g_t^2)$
> >
> > We would like to clarify that this is not the bridge we use. We follow the standard practice established in [1,2,3] of using a variance of $g_t^2 = \sigma_t^2 t (1 - t)$. Specifically, we use
> > $$P_{t|0,1}(x_t) = \mathcal{N}(x_t; t x_1 + (1 -t) x_0 + t (1 - t) \psi_{t,\eta}(x_0, x_1), \sigma_t^2 t (1-t))$$
> > in all experiments.
> >
> > > As I explained in my first review, a well-suited formulation would be to follow the setting of GSBM, which lets flexible the variance and learns it along the marginal mean.
> >
> > We politely disagree. Learning the variance is unnecessary in our setting where we consider a known reference stochasticity as part of the problem setup, as in this case the optimum of the leared variance would be equivalent to the the reference.
> >
> > > Following my first point, I am afraid that the comparison you propose with GSBM is not fair, especially related to the fact that you use the loss at Line 162, instead of their variational loss (6a) which is more theoretically grounded.
> >
> > We would like to clarify that the **loss (6a) in GSBM cannot take into account a non-zero reference drift directly**. Loss 6a only considers potential functions $V_t(x_t)$ and the kinetic energy of $u_t$, and therefore is theoretically problematic in our setting.  We therefore use our loss function, which we believe is a more fair comparison, although we are happy to include both versions in the updated draft of the paper. For completeness we also include a GSBM baseline with loss 6a found in the original paper of GSBM below—as the reviewer had originally requested. We find that CurlyFM is able to outperform both variations of GSBM on our oceans dataset across the three main quantitative metrics of interest. We further find our modification of GSBM’s loss allows it to better match the non-zero reference drift as evidenced by a lower cosine distance, while also performing slightly better in W1 distance.
> >
> > Table R4 Comparison of CurlyFM to GSBM on the Ocean currents dataset.
> > | | Cos. Dist. | L2 cost | W1 |
> > |-|-|-|-|
> > | **GSBM 15 (our loss)** | 0.075 ± 0.017 | 0.134 ± 0.030 | 0.248 ± 0.045 |
> > | **GSBM 15 (6a)** | 0.083 ± 0.037 | 0.134 ± 0.029 | 0.201 ± 0.058 |
> > | **CurlyFM**| 0.070 ± 0.001 | 0.107 ± 0.003 | 0.052 ± 0.004 |
> >
> >
> > We hope this helps alleviate the reviewer’s concern about our GSBM baseline. We are happy to clarify further if the reviewer has any final doubts. Please do let us know.

---

> > > ### Comment · Reviewer_9hEa · 2025-08-08
> > >
> > > Thank you for providing response to my concerns. Unfortunately, I still think that there are still misunderstandings related to the mathematical formulation of the bridge problem indeed. I'll try to precise exactly what bothers me.
> > >
> > > > We would like to clarify that this is not the bridge we use. We follow the standard practice established in [1,2,3] of using a variance of $g_t^2 = \sigma_t^2 t (1 - t)$. Specifically, we use $$P_{t|0,1}(x_t) = \mathcal{N}(x_t; t x_1 + (1 -t) x_0 + t (1 - t) \psi_{t,\eta}(x_0, x_1), \sigma_t^2 t (1-t))$$ in all experiments.
> > >
> > > Thank for detailing this part, which was not clearly stated before. From what I understand, given the elements presented in the paper and the first rebuttal,
> > > - $\sigma_t$ is the volatility of the target process
> > > - $g_t$ is the volatility of the reference process
> > > - you set $\sigma_t=g_t$, notably because it enables to correctly use the Girsanov formula in your setting (correct me if I am wrong) -> note that this is indeed the standard choice in references [1,2] mentioned by the authors.
> > >
> > > What I don't understand are the following :
> > > - you indicate in your last answer that you set $g_t^2 = \sigma_t^2 t (1 - t)$ thus matching references [1,2]. But as far as I understand, $g_t$ (the volatility of target and reference processes) is chosen constant in [1,2]. Is this an error of notation ?
> > > - assume that the previous point originates from an error of notation, ie assume that the volatility of the reference process $\mathbb{Q}$ is $g_t=\sigma$ (constant as in the experiments) and that you set the variance of the bridge approximating
> > > $Q_{t|0,1}$ as $\Sigma^2_t=\sigma^2 t (1 - t)$. In the case of the Brownian bridge (which is specifically the scope of [1,2]), I agree that $\Sigma^2_t$ is exactly the accurate variance : this can easily be obtained because Q_t|s   is tractable for $t>s$  (and Gaussian). Then one can use Bayes formula to get $Q_{t|0,1}$ (which is Gaussian). However, in
> > > the general case (ie $\mathbb{Q}$ has non linear drift),  $\mathbb{Q}_{t|s}$ is not tractable anymore (and probably not Gaussian), so the same thinking cannot be easily applied here. Nonetheless, you indicate in your last answer that :
> > >
> > > > Learning the variance is unnecessary in our setting where we consider a known reference stochasticity as part of the problem setup, as in this case the optimum of the leared variance would be equivalent to the the reference.
> > >
> > > Could you explain more in details how you are able to prove that the variance of the intractable bridge $\mathbb{Q}_{t|0,1}$ is indeed $\Sigma^2_t$ ? Maybe this is simple, but I don't see it. In the case where there may be a gap (ie $\Sigma^2_t$ is only an approximation of the true bridge variance), do you agree that it would be more legitimate to learn also the variance as in GSBM ?
> > >
> > > **About the variational loss of GSBM.** I agree with the authors on the fact that the settings of CurlyFM and GSBM are not exactly aligned. My concern was about the SDE/ODE mismatch in the loss $\mathcal{L}(\eta)$ (that I gave in my previous answer, but was not addressed), which does not appear in the GSBM loss (hence, not really related to the potential $V$). I will explain more in depth. Given the Gaussian Stochastic interpolant $X_{\eta,t}\sim Q_{\eta,t|0,1}$ (which approximates the ground truth bridge distribution $Q_{t|0,1}$), the goal of the first optimization step (in CurlyFM and GSBM) is the same : it is intended to find the parameters $\eta$ that enable to match the marginals of $\mathbb{Q}_t$. Two cases are possible :
> > > - (**flow matching**) either $\mathbb{Q}$ is induced by an ODE, ie $g_t=0$ (that is the default setting presented by the authors). In this case, the ODE with drift given by the conditional expectation $\mathbb{E}[\partial_t X_{\eta,t}|X_0,X_1]$ has the same marginals as $\mathbb{Q}$. Therefore, one can use the L2 loss $\eta \to \mathbb{E}[\|\partial_t X_{\eta,t}- f(X_{\eta,t})\|^2]$, with the expectation taken on $(X_0,X_1, X^{\eta}_t)$ (as properly done by the authors) to learn $\eta$.
> > > - (**bridge matching**) or  $\mathbb{Q}$ is induced by an SDE, ie $g_t=\sigma$ (constant). In this case, the marginal preserving SDE (which is targeted by authors, see Pb stated Line159) has the same volatility $g_t$ but drift given by $\mathbb{E}[\partial_t X_{\eta,t} +  a_t (X_{\eta,t} − µ_t)|X_0,X_1]$, where $\mu_t$ is the mean of $X_{\eta,t}$ and $a_t= \frac{1}{\Sigma_t}(\partial_t \Sigma_t -\sigma^2/2\Sigma_t)$ ($\Sigma_t^2$ : the variance of $X_{\eta,t}$).  Therefore, one can use the L2 loss $\eta \to \mathbb{E}[\|\\partial_t X_{\eta,t} +  a_t (X_{\eta,t} − µ_t)- f(X_{\eta,t})\|^2]$ to learn $\eta$. **My remark on GSBM was related to the additional term that is missing in your stochastic formulation**. Note : this is exactly the loss used by [1,2] in the tractable case (see App. H of DSBM paper).
> > > 
> > >
> > > Although these losses are the same as $\sigma \to 0$ (see App. A.1 of DSBM), I think the second loss should be considered when $\sigma\neq 0$.

---

> > ### Author Response · Authors · 2025-08-08
> > **Response to reviewer's latest response**
> >
> > We thank the reviewer for reviewing our previous response and for their useful suggestions on improving the clarity of bridge construction and GSBM baseline comparison to CurlyFM. We will include this in our updated manuscript. Here, we address the few remaining concerns raised by the reviewer.
> >
> > > you indicate in your last answer that you set $g_t^2 = \sigma_t^2 t (1-t)$  thus matching references [1,2]. But as far as I understand, the volatility of target and reference processes is chosen to be constant in [1,2]. Is this an error of notation?
> >
> > We now understand the reviewer's point regarding the notation, and we agree with the reviewer that this is an error in notation on our part. In particular, we set $g_t=\sigma_t=\sigma$ for some constant $\sigma$, and use $\Sigma_t^2 = \sigma^2 t (1 - t)$ as the variance of the bridge matching the set-up presented in [1,2]. We thank the reviewer for catching this, and we will update the notation used in our implementation to avoid any further confusion.
> >
> > > Could you explain more in details how you are able to prove that the variance of the intractable bridge  $\mathbb{Q}_{t| 0,1}$ is indeed $\Sigma_t^2$ ? Maybe this is simple, but I don't see it.
> >
> > The reviewer is indeed correct regarding this point and the fact that the variance of the general bridge is for a general non-linear reference drift is $\Sigma_t^2$. However, as stated very explicitly in our main paper we make the simplifying assumption and take the reference $\mathbb{Q}_{\eta, t}$ to be a **mixture of Brownian bridges** (stated on line 158) as a key modeling choice. In this specific case, all of our construction holds and we make no claims beyond this setting as it is not empirically needed for the application domains considered. We understand that this point may not have been transparent enough in the main paper and we will update the draft to include this more explicitly in the introduction and motivation part of section 3.
> >
> > > In the case where there may be a gap (ie $\Sigma_t^2$ is only an approximation of the true bridge variance), do you agree that it would be more legitimate to learn also the variance as in GSBM ?
> >
> > We would not describe it as more legitimate, as there are tradeoffs to the additional complication of learning the variance. However, we agree that there are situations where learning sigma is better as demonstrated in GSBM.
> >
> > ## About the variational loss of GSBM.
> >
> > Yes, we totally agree with the reviewer and now better understand the confusion. We follow the formulation of [3] which separately learns the deterministic flow and stochastic score functions, which then can be added together to calculate the stochastic drift or used separately to integrate deterministically. We note that we do this so that we can experiment with different stochasticity and simulate forward or backwards in time.
> >
> > For clarity we supply the python code we use to train and perform inference in CurlyFM. We hope this clears up the confusion around correctness. We note that here `score_model` is really the score scaled by `sigma^2 / 2`. This matches the correct conversion between ode drift and stochastic drift.
> >
> > **CurlyFM Training algorithm:**
> > ```python
> > for _ in range(n_iter):
> >     x0 = sample_x0(batch_size)
> >     x1 = sample_x1(batch_size)
> >     x0, x1 = coupling(x0, x1)
> >     t = torch.rand(batch_size).type_as(x0)
> >     eps = torch.randn_like(x0)
> >     lambda_t = (2* torch.sqrt(t * (1-t))) / sigma
> >     xt, xt_dot = get_xt_xt_dot(t, x0, x1, geodesic_model)
> >     xt = xt + eps * sigma
> >     vt = drift_model(xt, t)
> >     st = score_model(xt, t)
> >     flow_loss = torch.mean((vt - xt_dot) ** 2)
> >     score_loss = torch.mean((lambda_t * st + eps) ** 2)
> >     loss = flow_loss + score_loss
> > ```
> >
> > **CurlyFM Inference algorithm:**
> > ```python
> > for t in torch.linspace(0, 1, 100)[:-1]:
> >     stochastic_drift = drift_model(x,t) + score_model(x,t)
> >     x = stochastic_drift * dt + sigma * torch.sqrt(dt) * torch.randn_like(x)
> > ```
> >
> > To clarify this in the text, we will add the full pseudocode for learning the additional model for stochastic correction as well as the inference pseudocode for clarity.
> >
> > We hope this clears up any remaining misunderstandings in the construction of our bridge and comparison to the GSBM baseline. We are more than happy to clear up any additional questions the reviewer may have. Please do let us know.
> >
> > ## References
> > [1] Shi, Yuyang, et al. "Diffusion schrödinger bridge matching." NeurIPS (2023)
> >
> > [2] Peluchetti, Stefano. "Diffusion bridge mixture transports, Schrödinger bridge problems, and generative modeling." JMLR (2023)
> >
> > [3] Tong, Alexander, et al. "Simulation-free Schrodinger bridges via score and flow matching." AISTATS (2024).

---

> > > ### Comment · Reviewer_9hEa · 2025-08-09
> > > **Final answer**
> > >
> > > Thank you for acknowledging these elements about the bridge matching formulation, which make plenty sense at this stage. Overall, the authors have addressed the main concerns I had about the paper, that is (i) the lack of mathematical details to establish the loss in the deterministic case (what is presented in the main paper) and (ii) the extension to the stochastic case (which I recall is the canonical setting of the SB problem), that was missing in the paper. I deeply thank the authors for their involvement in the discussion, which enabled to bring light on ambiguities and imprecisions.
> > >
> > > With the elements given by the authors throughout this discussion, I am inclined to increase my score if they are integrated to the updated manuscript (as advocated by the authors). I will recall them for full clarity:
> > > -  a detailed explanation about the experimental setting that is considered : deterministic/stochastic ? value of volatility coefficients ? link with the generalized SB formulation ?
> > > - a detailed presentation of the assumptions made throughout the paper : modeling of the Bridge (for instance, which does not include flexible variance in the stochastic case + choice of the variance instead), assumptions made on the reference drift to justify the use of kernels, assumptions to compute the cost of OT plan...
> > > - a rewriting of Section 3.2 to give more details about the losses that are introduced : I have the feeling that for non expert readers, it is difficult to understand where these elements come from, what are their limitations. For instance, the subtle difference between the bridge matching (see last author's response) and the flow matching (in the main) frameworks are not mentioned. It would definitely benefit to the paper to spend more time on introducing the items with details (i) approximation of the bridge (for both deterministic and stochastic cases), (ii) computation of the target SDE -> as commonly presented in bridge matching papers. Following the last author's response, the elements related to the stochastic formulation should appear.
> > > - a comparison with baselines in deterministic (-> OT-CFM) and stochastic (-> SB-CFM, SF2M) should be provided, along an ablation study of the effect of hyperparameter $\sigma_t$
> > > - an in-depth discussion about the comparison with GSBM paper, which has the same spirit : solving a SB problem where the reference drift does not have an analytical formula when taking the bridge + 2-step procedure implying (i) and (ii) (see above for description of these steps). This discussion should include the main differences : iterative vs non iterative + choice of the reference process (tilting of Brownian motion for GSBM, complex drift that can only be evaluated at some points for CurlyFM).
> > >
> > > If all these changes are well integrated, I think that the resulting paper would be of high quality. In this direction, I accept to increase my score.

---

> > > > ### Author Response · Authors · 2025-08-09
> > > > **Response to reviewer's final answer**
> > > >
> > > > We are deeply grateful to the reviewer for their time, dedication and engagement during this rebuttal process. We believe the additional discussions and resulting clarifications, new experiments, and ablations have strengthened our overall paper. We are also pleased to hear that the reviewer is positive about increasing their score upon making the suggested changes in the updated draft. We now briefly review our proposed changes in light of this insightful rebuttal discussion with the reviewer in an effort to solidify the reviewer's positive stance:
> > > >
> > > > **1. Updates to the experimental setting**: We will provide additional details on which aspects of the experimental setting correspond to the deterministic limit we considered in the original paper. We will also include the new ablations provided in this rebuttal for the stochastic case including the exact value of the volatilities used in the appendix. We will further update the notation based on the typo’s that reviewer helped us catch and particularly making it clear that we follow the setting in [1,2,3] as well as fixing the notation on $g_t$ and $\sigma_t$ as outlined in our previous response.
> > > >
> > > > **2. Details on all all assumptions**: We will additionally update the paper to discuss the exact assumptions used in the problem formulation. These include the further emphasis on the modeling assumption of the marginals of the reference process being a mixture of Brownian bridges. We will further make clear that the Bridge we use in the stochastic case uses a fixed variance and provide experimental details. Finally, we also include the clarifications about the assumptions on the kernels and the estimate of the OT cost we use with justifications.
> > > >
> > > > **3. Rewrite of section 3.2**: We thank the reviewer for this suggestion. We agree that section 3.2 is currently terse and difficult to parse for non-experts. We will update the paper to clearly define the deterministic flow matching case and the more general approximation to Bridges that we clarified in the rebuttal including the other technical assumptions we made along the way.
> > > >
> > > > **4. Baselines for OT-CFM and SB-CFM, SF2M**: We will include these additional baselines in the updated paper for the rebuttal experiments which also include an ablation on various hyperparameters including dimension and stochasticity in the toy setting. We will include rebuttal experiments and make clear distinction between deterministic and stochastic settings as well as clearly state $g_t$ and $\sigma_t$ used in the experiments.
> > > >
> > > > **5. GSBM**: We thank the reviewer for pushing to include the GSBM baseline. We will include this baseline and the corresponding ablations in the updated draft and also include a discussion on the main differences between two set-ups. In particular, we will focus on the iterative vs non-iterative nature of the two models and their ability to learn complex drift, and the choice of reference process used. As per our previous response, we will also include the python pseudo-code to train and perform inference in CurlyFM, to aid clarity on assumptions in our comparison to GSBM baseline.
> > > >
> > > > ## Concluding Remarks
> > > > We again deeply thank the reviewer for their strong engagement and extensive discussion during the rebuttal period. We are glad to hear that the provided responses cleared the chief concerns and that the reviewer is positive about increasing their score. We thank the reviewer again for their useful suggestions in their final answer and we will reflect these in the updated manuscript.
> > > >
> > > > ## References
> > > > [1] Shi, Yuyang, et al. "Diffusion schrödinger bridge matching." NeurIPS (2023)
> > > >
> > > > [2] Peluchetti, Stefano. "Diffusion bridge mixture transports, Schrödinger bridge problems and generative modeling." JMLR (2023)
> > > >
> > > > [3] Tong, Alexander, et al. "Simulation-free schrodinger bridges via score and flow matching." AISTATS (2024).

---

### Official Review · Reviewer_BnED · 2025-07-02

**Clarity:** 3
**Significance:** 3
**Originality:** 3
**Rating:** 5
**Confidence:** 3

**Summary:**

An approach is presented that allows learning flows that include drifts rather than only gradient-field dynamics. The authors first learn an interpolant (neural path) and then fit a velocity field based on the coupling provided by the interpolant.

**Questions:**

- It is unclear how the trade off between matching the drift and marginals is handled. What knob needs to be turned in Algorithm 1 and 2 to trade off these two learning objectives? A more detailed discussion should be added because this question comes up several times in the paper (e.g., Intro on page 3 top and line 186-187)

- There are other approaches than simply pairwise connecting marginals with OT such as action matching by [Neklyudov et al., 2024]. These approaches incur often lower training costs than solving many OT problems. Can this be used as a first stage in the presented approach as well to avoid some of the limitations of pairwise interpolants?

- Minor questions: The authors state that TrajectoryNet is much slower and unstable (line 183). Can the authors quantify this for one of the examples?

**Ethical Concerns:**

["NO or VERY MINOR ethics concerns only"]

**Final Justification:**

There were no major issues before the rebuttal. The answers by the reviewers clarified some minor comments and overall confirmed by Accept rating.

**Limitations:**

Yes, the authors have addressed in the Conclusion the limitation of first learning the reference field, which is the main limitation of this approach.

**Quality:**

3

**Strengths And Weaknesses:**

Strengths:
+ Learning non-gradient field dynamics is indeed an important problem and little work in this direction has been done. So this work is timely and useful.

+ The proposed two-stage approach of first learning a neural path interpolant and then fitting a velocity field is straightforward but reasonable and sound. The approach is well described.

+ There are extensive numerical experiments that demonstrate that the approach captures drift dynamics whereas standard conditional flow matching fails

Weaknesses:
- There are extra training costs incurred by the additional steps (e.g., the OT step) that can be high. No discussion about it is provided.

- There is a limitation by first fitting the neural path interpolant, which is not addressed well. For example, if the initial neural path is poorly fitted, how will it affect the second step? The authors do mention briefly in the Conclusion that the approach hinges on inferring the reference field well, but this requires more discussion.

---

> ### Author Rebuttal · Authors · 2025-07-31
>
> We are grateful to the reviewer for their time and effort reviewing our paper, as well as their positive appraisal of our work. We appreciate that the reviewer agrees with us that learning non-gradient field dynamics “is indeed an important problem and little work in this direction has been done”. We are also heartened to hear that the reviewer finds our two-stage approach to be “reasonable and sound” and that our approach is “well described.” Finally, we thank the reviewer for finding our set of numerical experiments “extensive” and that they demonstrate settings where CurlyFM captures drift dynamics while “standard conditional flow matching fails”. We next answer the main questions raised in the review.
>
> ## Tradeoff between matching the drift and marginals
>
> We acknowledge the reviewer's question regarding the possibility of an inherent trade-off in matching the reference drift versus matching the marginals. We believe there may be a potential cause for confusion, as there is no single hyperparameter that negotiates a trade-off between Algo 1 and Algo 2. Indeed, CurlyFM is a two-stage method which first (1) matches the reference drift as closely as possible, and then (2) uses the learned reference drift to learn the optimal marginals that solve the mass transport problem. The hyperparameter and simulation-free nature of CurlyFM is a significant advantage over other methods, e.g., TrajectoryNet [1].
>
> Our comments on the top of page 3 and lines 186-187 were simply to illustrate the fact that learning the reference drift perfectly **does not necessarily solve the mass transport problem as a Schrödinger Bridge**, and this is exactly the reason we need Algo 2. As a result, the perfect final model should be a valid Schrödinger Bridge, which is close to the reference as much as is required to actually solve the mass transport problem and no more. We understand that this point may not have been sufficiently clear in the main paper, and we will update the draft to include more commentary on this point.
>
> ## Other mechanisms for connecting marginals
>
> We thank the reviewer for their interesting question regarding alternative approaches to connecting the marginals without solving an OT problem, such as Action Matching. Unfortunately, to the best of our knowledge, methods such as Action Matching are more computationally expensive than what is presented in Algo 2. This is because Algo 2 remains simulation-free due to an unbiased stochastic estimate of the OT cost—rather than computing the entire cost using simulation. In practice, this is quite fast and is not a significant computational overhead in all the experimental settings we consider in the paper. At a broader level, approaches that do not connect marginals via OT run the risk of not being valid solutions to the Schrödinger Bridge problem, which is the thesis of our work. We hope that this answers the very interesting question raised by the reviewer.
>
> ## Computational speed of TrajectoryNet
>
> Thank you for your great suggestion. We have now included the computational cost of TrajectoryNet (and Curly-FM) in Table R1. We report these results for the 2D and 10D settings for the experiments reported in Table 4.
>
> Table R1: Computational efficiency comparison between CurlyFM and Trajectory Net
> | Method           |        d = 2 (seconds)    |        d = 10 (seconds)   |
> |--------------------|-------------------------------|-------------------------------|
> | TrajectoryNet  |    $17005 \pm 110$     | *$43080 \pm 65$         |
> | CurlyFM         |     $\mathbf{1429 \pm 31}$        |  $\mathbf{2471 \pm 10}$           |
>
>
> As observed in the above table, we find that TrajectoryNet is considerably slower than CurlyFM and is not a scalable approach. This is evident in that training TrajectoryNet takes, on average, 11 times longer than Curly-FM in the 2D case, and 17 times longer in the 10D case. Moreover, as dimensionality increases from 2D to 10D, Curly-FM incurs an increase in computational cost by ~1.7x, whereas TrajectoryNet incurs at least a 2.5x increase in computational cost. We use (*) to denote runs that did not finish within allotted resource allocation time. We will include these results in the updated draft of the paper.
>
> ## Concluding remarks
>
> We thank the reviewer again for their time and effort during this review process. We hope that our rebuttal here was successful in answering all the great questions raised by the reviewer and allows the reviewer to continue positively endorsing our paper. We are also happy to answer any further questions the reviewer may have during the remainder of the rebuttal period.
>
>
> ## References
> [1] Tong, Alexander, et al. "Trajectorynet: A dynamic optimal transport network for modeling cellular dynamics." International conference on machine learning. PMLR, 2020.

---

> > ### Comment · Reviewer_BnED · 2025-08-03
> >
> > I thank the authors for their detailed comments. I don't have any further questions and I maintain my Accept rating.

---

### Official Review · Reviewer_Tr4n · 2025-07-03

**Clarity:** 2
**Significance:** 2
**Originality:** 2
**Rating:** 4
**Confidence:** 3

**Summary:**

The paper introduces Curly Flow Matching (Curly-FM), a simulation-free method for learning non-gradient, possibly periodic dynamics from snapshot data and approximate velocity measurements.  The authors cast trajectory inference as a Schrödinger-bridge problem whose reference process has non-zero drift built from the given velocities.  Because (most of the) existing diffusion/flow-matching techniques assume zero drift, the paper proposes a new method to account for non-zero drift via a two-stage approximation: (1) find a neural path interpolant such that its instantaneous velocity matches the reference drift, and (2) marginal flow matching to learn a drift that marginally modifies the neural path interpolant to make sure that the induced stochastic differential equation satisfies the marginal constraints at the initial and final times. The authors show promising results on empirical and real data experiments with rotational flows, improving over state-of-the-art baselines.

**Questions:**

In this section, I will get in more details of the comments provided in the strengths and weaknesses section above, as well as provide some additional minor comments and questions.

**Major Comments and Questions**

- When I first encounter $\eta$, it is not clear what this represents and why we need it. I think it would be great if the notation is setup and clearly described why we need this before using it. Similarly (and possibly relatedly) I am not sure I follow how $Q_{\eta}$ (line 158) is actually constructed. I get it that is parametrized as a mixture of Brownian bridges, but above you are saying that these bridges need not be Brownian (Line 154). Is this making a modeling/approximation choice? I think it would be much clearer if the mixture were explicitly written out, with each component described both formally and intuitively.
- After line 159, $f*_{\eta}$ seems to correspond to the minimum value of the integral. Is that correct? Or is it instead meant to refer to one of the $f_{t,\eta}$ terms? I am also not sure that the star notation is used anywhere later in the paper, so I don't follow why we need this.
- What is the role of $\kappa$, and how is it chosen in practice? Is it the same for each experiment? Why is it introduced below line 162? Similarly, what is the precise function of the neural path interpolant? How does it relate to model drift and the approximated velocity field? My understanding is that with the neural path interpolant I am trying to make the path $x_{t,\eta}$ curly, and I am doing it by finding the best interpolant that matches a reference velocity field. And then once I found the interpolant I can use it in the derivative of the path to approximate the drift in the original SDE (equation (3)) with the loss after Line 172.
- Overall, I find Section 3.1 quite difficult to follow, as it lacks both formal clarity and intuitive explanation. A clear breakdown of the two main steps, along with an explanation of what each quantity represents (both formally and intuitively), would greatly improve readability. I understand the high-level two-step procedure from the introduction, but I find it challenging to connect that to the formalism presented in this section.
- More broadly, I think I am confused by how the reference velocity field is obtained. Is it computed by deriving the velocity along the stochastic path interpolant and then fitting a neural network to that data, using the loss after equation 162? Or are the velocities approximated using finite differences outside the optimization loop and then used in a regression setup? Is approximating the velocity field necessary at all? Could the method be adapted to scenarios where the true velocity field is known? And how does the accuracy of this approximation affect the performance or theoretical guarantees of the method? I know these are a lot of questions, but I am asking because I think this is the crucial innovation of the method and I am wondering whether it can be spelled out better. I am also asking because for most of these data there is no notion of observed trajectory for a particle, since the particles are unpaired, so I am not sure if this approach is impacted by this or not.
- Lines 265–267: What is meant here? Could you provide a citation or further explanation? This relates to my broader confusion about the role of the velocity field (as noted above). It’s also not clear whether the method assumes access to RNA velocity fields for this experiment, as seems implied in Line 230.
- How does the method generalize to more than two time points? In the ocean current experiment, there are more than two time points.  But in the method part the discussion is focused on just two end points at time 0 and 1. But from the experiment results it doesn't look like you're learning the drift piecewise between each pair of points (since the connections look smooth enough), so what is exactly going on? Does the need to approximate multiple velocity fields introduce cumulative bias?

---

**Minor Comments and Questions**

- I really like the inclusion of Figure 1 on the first page! Could you increase the font size for the legend labels? Also, it would be helpful to spell out “OT-CFM” and “CFM” in the figure caption, since the acronyms are not introduced in the main text at that point. Many readers will look at the figure together with the abstract and will not yet know what those terms mean.
- Line 43: I suggest adding a “Most of” qualifier here, since, as noted above and in the extended related work section, there are existing methods that can model non-gradient dynamics.
- Line 68: The meaning of this sentence is unclear — could you rephrase or clarify?
- Lines 140–141: A citation is needed.
- Equation (9): Unless this particular characterization of the Schrödinger Bridge problem is used later in the paper, it may not be necessary to include it here.
- Line 146: The phrase “We approximate...” makes it sound like these are the algorithms used in the current paper, which is not true. A more general phrasing would be preferable.
- Lines 152–153: Is there a double negation issue here? It may be worth rephrasing for clarity.
- Line 170: Is Equation (10) the correct reference? If so, what exactly do you mean by “total cost” in this context?
- It would be helpful to explain how boldface is used in the tables. Currently, it seems that only the proposed method is bolded when it achieves the best performance, while other methods are not highlighted even when they perform best. This could be misleading unless explicitly stated.
- Not a big deal, but note that Figure 5 currently appears before Figures 3 and 4.

My score is mostly driven by the points raised in the main comments and weaknesses sections. I will be very happy to revise my score if the authors can clarify my confusion and doubts.

**Ethical Concerns:**

["NO or VERY MINOR ethics concerns only"]

**Final Justification:**

As explained in my reviews, I found the paper an interesting submission tackling an important problem. I had some initial concerns (raised in my review) and I appreciated the clarifications and the authors’ efforts to address them. I also appreciated the follow-up comment. For these reasons, I increased my score.

**Limitations:**

Yes, the authors discuss some limitations of their work in the discussion section. In particular, they briefly note that the accuracy of the inference process depends on how well the reference family can be reconstructed. As mentioned above, I believe it would be very valuable to explore this more formally — for example, by investigating whether certain assumptions on the data or on the quality of the derivative approximation could guarantee a desired level of inference accuracy.

**Quality:**

3

**Strengths And Weaknesses:**

**Quality.** The submission is generally technically sound. It does not contain any formal propositions or main theoretical results, but most of the claims are well supported by the experimental findings. The authors do a good job discussing important background ideas, although in some parts it would be helpful to include either more intuitive explanations or more precise references (see the *Questions* section below for details).

**Clarity.** I find most of the paper to be clearly written. I appreciate that the authors clearly motivate their work and set up the problem before presenting potential solutions. However, the paper becomes harder to follow in Section 3.1, where additional details and intuition would be very helpful for the reader to fully understand the method and be able to reproduce it. I will provide more specific feedback in the *Questions* section, but in my view, this is the first major weakness of the paper.

**Significance.** The results have the potential to be impactful for the community. I believe that some researchers will find the experimental contributions appealing and may be motivated to pursue theoretical developments based on them. Additionally, practitioners working with similar data may be interested in applying this method. That said, the paper would benefit from an analysis of how crucial it is to accurately approximate — or have access to — the velocity field in order to obtain good results, since this data may not be readily available in all practical settings. More broadly, I think the paper should include a more thorough discussion of the role of the velocity field in the proposed approach.

**Originality.** The paper presents a new method that clearly differs from most of the prior works cited. However, the authors repeatedly claim (e.g., Lines 43 and 151) that existing methods can only model gradient field dynamics, which is not accurate in general. For example, the work by Shen et al. (2024), which the authors use in one of their experiments, supports more general dynamics, The two works by Bartosh et al. (2024 and 2025) are also relevant in this context—they are briefly mentioned in Section 5 (line 284), but the authors do not provide any intuition about how these methods differ from their own (beyond a short comment about the “simulation-free” aspect) and do not compare against them experimentally. I view this as the second major weakness of the paper. If the proposed method is very similar to Bartosh’s work and its main advantage is being simulation-free, then I would expect some discussion or analysis of computational speed or efficiency. Without such analysis, it is difficult to justify a new method that does not clearly differentiate itself from recent work on a very similar task. Overall, I believe the work is original and constitutes an interesting contribution, but the discussion of related work should be expanded and the motivation for the proposed approach strengthened.

---

> ### Author Rebuttal · Authors · 2025-07-31
>
> We thank the reviewer for their time and effort in reviewing our paper. We appreciate that the reviewer found our paper “technically sound” and that the majority of our claims “are well supported by the experimental findings,” which have the potential to be impactful to the community. We are also heartened to hear that the reviewer finds our paper to be “clearly written” and that the work is clearly motivated. Finally, we thank the reviewer for acknowledging that CurlyFM “presents a new method that clearly differs from most of the prior works”. We now address the key points of clarification raised in the review, grouped by theme.
>
> ## Questions regarding the clarity of presentation
>
> We value the reviewers' feedback on improving the presentation clarity of our paper. We now answer the specific pain points raised in the questions.
>
> > When I first encounter $\eta$, it is not clear what this represents …
>
> $\eta$ represents a general mixture of conditional bridges $\mathbb{Q}_{\eta, t}$. As a modelling choice, we then set  $\eta$ as the parametrized neural path interpolant, which we use to parametrize **a mixture of Brownian bridges** as stated in line 154. We understand that this point may not have been clear, and we will update the notation to improve clarity.
>
> > After line 159, $f^\star_\eta$ seems to correspond to the minimum value of the integral. Is that correct? …
>
> Indeed, this is the correct interpretation. We use the star notation to denote the solution to an optimization problem, such as $f^\star$ or the optimal coupling $\pi^*$, which leads to the Schrödinger Bridge.
>
> > What is the role of $\kappa$
>
> The kernel allows us to construct the reference process everywhere in space and time by smoothening out ground truth instances of the reference drift found in the data samples from each time marginal. This is part of the problem setup and is considered given.
>
> > what is the precise function of the neural path interpolant?
>
> The neural path interpolant is used to match the reference drift $f_t$, which in general is no longer a straight path between $x_0$ and $x_1$. Intuitively, this makes our paths bend and can be thought of as the mean of the conditional mixture of Brownian bridges. To select which $x_0$ is paired with $x_1$ we use the neural path interpolant as part of Algorithm 2 to compute the OT cost and consequently solve the SB problem.
>
> ## Differences with related work (e.g. Shen et. al 2024 and Bartosh et. al (2024 and 2025))
>
> ### Shen et. al 2024
>
> Shen et. al 2024 use an iterative bi-level algorithm wherein step 1) they solve a Schrödinger bridge and estimate the forward-backward drift given a current guess of the reference drift, 2) Given the solution to step 1, they simulate trajectories to estimate a new best reference guess. This bi-level approach is considered since the reference drift belongs to a family of possible reference drifts rather than a single prescribed one. In contrast, CurlyFM employs a 2-stage algorithm that is not iterative and also does not search over a family of reference drifts. More critically, CurlyFM is also simulation-free, which is achieved by making the modeling assumption that the conditional mixture of bridges is modelled as Brownian bridges. In practice, this means that CurlyFM is significantly more efficient to train. We report in Table R1 the computational cost in wall clock time for each approach and notice that CurlyFM completes the Ocean currents problem in a matter of minutes with better performance (see Table 2 in main paper), while the method proposed in Shen et. al 2024 is in the order of multiple hours— thus clearly demonstrating the computational benefit and empirical caliber of CurlyFM. Finally, note, both Shen et. al 2024 and CurlyFM **can model non-gradient field dynamics**, but only the latter is simulation-free.
>
> Table R1: Compute cost in hours
> | Method                   | Hours |
> |--------------------------|--------|
> | DM-SB                   | 15.44  |
> | Vanilla-SB              |  0.43  |
> | TrajectoryNet           |  7.44  |
> | SBIRR (Shen et al. 2024)|  4.67  |
> | CurlyFM (Ours)          |  **0.06**  |
>
>
>
> ### Bartosh et. al (2024 and 2025)
>
> Both papers from Bartosh et. al consider Latent SDEs with simulation-free training. The key difference between these papers and CurlyFM is that they do not solve a Schrödinger Bridge problem as considered in our work. Furthermore, SDEs in CurlyFM are not Latent SDEs. More precisely, Latent SDEs learned in Bartosh et. al 2025, have no theoretical basis to converge to a Schrödinger Bridge—i.e. the learning process does not minimize the KL between a reference process, and they do not learn a bridge in their current presentation of the paper. Consequently, we argue that the approaches and goals of these papers are different from CurlyFM and cannot be meaningfully compared experimentally, despite both approaches learning an SDE in a simulation-free manner. Whilst it may be possible to build a computational approach to solving the SB problem with latent SDEs, this is an orthogonal research direction to this paper. We will revise our related work section to include a more detailed commentary.
>
> ## Handling more than two time points?
>
> We acknowledge the reviewer's question. Please note we handle the multi-marginal case as done in prior work (e.g., MFM [1]), where by we construct CurlyFM between two time marginals by skipping an intermediate time marginal, for example, $t=1$ is skipped when $t=0$ and $t=2$ data is used. After which we proceed sequentially by constructing CurlyFM (instantiated from the previous iteration) on the next two time marginals and skipping the next time marginal, i.e., $t=2$ is skipped and $t=1$ and $t=3$ are used.
>
> ## Access and role of the reference velocity field
>
> In our considered setting, we assume access to the ground truth reference drift at samples $x_0 \sim \mu_0$ and $x_1 \sim x_1$ as part of the problem setup. Note that in the high-impact application domains we consider, such as trajectory inference in single-cell data, we have access to the RNA velocity [2,3]. As we need to estimate CurlyFM everywhere in space and time during training, we construct a smoothed version of the reference drift by using a kernel $\kappa_t$. This allows us to construct the reference drift $f_t (\mathbf{X}t)$---using knowledge from the ground truth reference drift in the existing dataset— in places where there are no ground truth samples. Consequently, we take $f_t$ in these intermediate points as our ground truth reference drift.
>
> > Could the method be adapted to scenarios where the true velocity field is known?
>
> Yes! If the reference is known everywhere, we would simply plug it in for $f_t$ rather than use our smoothed version via $\kappa_t$.
>
> > How does the accuracy of this approximation affect the performance
>
> This is a great question. To ablate this, we now construct a noisy reference drift for various noise levels in the tables below (for the Ocean Currents dataset). The noisy reference drift $f_{t}^{\text{noise}}$ is obtained as a linear combination of the ground truth reference drift $f_{t}$ and noise from a standard gaussian distribution ($ f_{t}^{\text{noise}} = (1 - \beta) * f_{t} + \beta * \text{noise}$). We find that the performance between $\beta = 0$ (no noise, i.e. regular CurlyFM) and $\beta = 0.25$ (25\% noise) are similar whilst the performance for $\beta = 0.5$ and higher gradually becomes worse, as the noise dominates over the ground truth reference drift in $f_{t}^{\text{noise}}$. This shows that CurlyFM is robust to moderate amounts of noise added to the ground truth reference drift.
>
> We hope that our answers and ablation in Table R2 help alleviate the reviewers' concerns. We will also include this ablation in the updated draft of the manuscript.
>
> Table R2: Ablation with noisy reference drifts
> | $\beta$ | Cos. Dist.     | L2 cost       | W1            |
> |--------:|----------------|---------------|---------------|
> | 0.00    | 0.062 ± 0.003  | 0.143 ± 0.010 | 0.034 ± 0.006 |
> | 0.25    | 0.057 ± 0.033  | 0.021 ± 0.036 | 0.051 ± 0.030 |
> | 0.50    | 0.087 ± 0.047  | 0.301 ± 0.085 | 0.091 ± 0.046 |
> | 0.75    | 0.261 ± 0.123  | 0.381 ± 0.120 | 0.145 ± 0.062 |
> | 1.00    | 0.428 ± 0.157  | 0.445 ± 0.121 | 0.237 ± 0.079 |
>
>
> ## Concluding remarks
>
> We thank the reviewer for their time and effort in reviewing our work, and we hope our efforts to clarify the main points, along with the additional ablations, allow the reviewer to consider improving their score, as they initially said they would be open to. We are also more than happy to answer any other questions that arise.
>
> ## References
>
> [1] Kapusniak, Kacper, et al. "Metric flow matching for smooth interpolations on the data manifold." Advances in Neural Information Processing Systems 37 (2024): 135011-135042.
>
> [2] A. Riba, A. Oravecz, M. Durik, S. Jiménez, V. Alunni, M. Cerciat, M. Jung, C. Keime, W. M. Keyes, and N. Molina. Cell cycle gene regulation dynamics revealed by rna velocity and deep-learning. Nature communications, 13(1):2865, 2022
>
> [3] V. Bergen, M. Lange, S. Peidli, F. A. Wolf, and F. J. Theis. Generalizing rna velocity to transient cell states through dynamical modeling. Nature biotechnology, 38(12):1408–1414, 2020.

---

> > ### Comment · Reviewer_Tr4n · 2025-08-06
> >
> > Thank you for the detailed and thoughtful rebuttal. I believe the authors did a great job addressing many of the concerns, and I really appreciated both the added ablation on the velocity field and the clarifications on related work and presentation. I think I now have a better sense of the method's strengths, and overall I believe the paper will improve with the promised changes. That being said, I want to reiterate that the method section — particularly Section 3.1 — would really benefit from significant rewriting to improve clarity and accessibility. I appreciate the authors trying to explain in the rebuttal parts of this, and I strongly encourage them to add the same level of details (but even more in some sense) to the revised version of the paper. Finally, I’m still unclear on the multi-marginal case: once you have used t=0 and t=2 and then t=1 and t=3, how do you concatenate these two bits? I am sure I am missing something simple here, but it would be great if you could further clarify. In summary, this is a promising and original contribution, and the authors did a great job addressing my comments, so I am raising my score accordingly.

---

> > > ### Author Response · Authors · 2025-08-06
> > > **Response to reviewer response**
> > >
> > > We thank the reviewer for their time and effort in reviewing our work. We are pleased that the reviewer found our work to be a promising and original contribution in the field and for their decision to raise their initial rating—we greatly appreciate it! We appreciate reviewer’s useful suggestions to improve clarity in the theoretical portion of the paper such as Section 3.1 and we will include these as well as the ablation on velocity field in the updated manuscript. We now answer the remaining question raised by the viewer.
> > > ## Multi-marginal case
> > > In the multi-marginal setting, we train by randomly sampling interpolation times $t$ within intervals corresponding to each adjacent pair of marginals—for example these intervals will be [0,1] and [1,2] if our marginals lie at $t=\\{0,1,2\\}$. For each interval $[t_i, t_{i+1}]$, we sample a random time $t$, then compute both the neural interpolant $x_t$ and its derivative $\dot{x}_t$. We also compute global time $\hat{t} = t_i + t$, if say t=0.5, the global time $\hat{t}$ for the second marginal pair will be $\hat{t} = 1 + 0.5 = 1.5$. This effectively ensures that neural interpolant and its derivative match the global axis of time. We then connect all the marginals by concatenating neural interpolant, its derivative and global times into single tensors and use these as inputs to train our neural path interpolant in algorithm 1 and the drift in algorithm 2.

---

### Note · Authors · 2025-08-13

Dear AC, SAC, and Reviewers,

As the rebuttal discussion concludes, we thank you for steering a productive process and for reviewers’ constructive questions and suggestions. We summarize outcomes below.

**Reviewer Tr4n** The reviewer raised questions about clarity, the role of the kernel and the reference-drift approximation, the multi-marginal setup, and comparisons to the Bartosh et al and Shen et al baselines. We added an ablation that perturbs reference velocities with Gaussian noise, clarified differences between CurlyFM and baselines, and detailed our assumptions for bridge construction, training, and connecting marginals. The reviewer concluded: “the authors did a great job addressing my comments, so I am raising my score accordingly.”

**Reviewer BnED** We provided theoretical clarifications on matching drift and marginals, expanded comparison of TrajectoryNet and CurlyFM, and added computational cost comparisons. The reviewer wrote: “I don't have any further questions and I maintain my Accept rating.”

**Reviewer 9hEa** We had an in-depth discussion with the reviewer who provided very high-quality feedback, suggested additional baselines, a new toy example to hyperstress CurlyFM, additional experiments in the stochastic setting, and further clarifications. We provided new experimental results for CurlyFM with ablations on stochasticity and dimension, and comparison to GSBM, SF2M, OT-CFM and SB-CFM baselines. We had extensive discussion on our bridge construction and use of kernels, which we clarified during rebuttal. In their final answer, the reviewer suggested a useful list of changes to the manuscript, which we will include. The reviewer wrote: “If all these changes are well integrated, the resulting paper would be of high quality. In this direction, I accept to increase my score.”

**Reviewer qrFn** We clarified approximations in SB construction, the link between RNA velocities and SDE drift, and the impact of kernel approximation. The reviewer agreed this is a reasonable modeling choice and asked that the approximation be stated explicitly, adding: “As my score is already positive, I will maintain it.”

We are grateful to the AC and the reviewers for their time and dedication during the rebuttal period and we hope the AC considers the detailed history of this discussion in their final assessment.

---

### Decision · Program_Chairs · 2025-09-17

**Decision:**

Accept (poster)

**Comment:**

This paper proposes a two-stage method for learning non-gradient field dynamics. Authors appreciated the novelty of the and significance of the method. The main remaining issue were concerns regarding clarity. In particular, reviewers Tr4n and 9hEa bring up some significant issues of clarity / opportunities for in the manuscript. The authors have committed to make these changes. These are quite substantial, to the degree that I consider it borderline if it might be appropriate for the paper to undergo another round of review. But I am trusting the authors to make the promised changes.